# Efficient Split-Mix Federated Learning for On-Demand and In-Situ Customization

**Junyuan Hong[1], Haotao Wang[2], Zhangyang Wang[2] and Jiayu Zhou[1]**

[1]Department of Computer Science and Engineering, Michigan State University
[2]Department of Electrical and Computer Engineering, University of Texas at Austin
`{hongju12,jiayuz}@msu.edu, {htwang,atlaswang}@utexas.edu`

## Abstract

Federated learning (FL) provides a distributed learning framework for multiple participants to collaborate learning without sharing raw data. In many practical FL scenarios, participants have heterogeneous resources due to disparities in hardware and inference dynamics that require quickly loading models of different sizes and levels of robustness. The heterogeneity and dynamics together impose significant challenges to existing FL approaches and thus greatly limit FL's applicability. In this paper, we propose a novel Split-Mix FL strategy for heterogeneous participants that, once training is done, provides *in-situ customization* of model sizes and robustness. Specifically, we achieve customization by learning a set of base sub-networks of different sizes and robustness levels, which are later aggregated on-demand according to inference requirements. This split-mix strategy achieves customization with high efficiency in communication, storage, and inference. Extensive experiments demonstrate that our method provides better in-situ customization than the existing heterogeneous-architecture FL methods. Codes and pre-trained models are available: `https://github.com/illidanlab/SplitMix`.

## 1 Introduction

Federated learning (FL) (Konečný et al., 2015) is a distributed learning paradigm that leverages data from remote participants and aggregates their knowledge without requiring their raw data to be transferred to a central server, thereby largely reducing the concerns from data security and privacy. FedAvg (McMahan et al., 2017) is among the most popular federated instantiations, which aggregates knowledge by averaging models uploaded from different participants.

When deploying federated learning, one challenge in real-world applications is the run-time (i.e., test-time) *dynamics*: The requirements on model properties (e.g., inference efficiency, robustness, etc.) can be constantly changing during the run-time, depending on the status of the devices or the outside environment. One common and specific type of dynamics is *resource dynamics*: For each application, the allocated on-device resources (e.g., run-time memory, CPU bandwidth, etc.) may vary drastically during run-time, depending on how the resource allocation of the running programs are prioritized on a participant's device (Xu et al., 2021). Another type of dynamics is the *robustness dynamics*: The constantly changing outside environment can make different requirements on the safety (or robustness) level of the model (Wang et al., 2020). For instance, the quality of real-time videos captured by autonomous cars can suddenly degrade, e.g., on entering a poor-lighted alley or tunnel from a well-lighted avenue, on entering a section of bumpy road which leads to a sudden burst of blurring in the videos, etc. In such cases, a more robust model should be quickly switch in and replace the one used on benign conditions, in order to prevent catastrophic accidents caused by wrong recognition under poor visual conditions. Such dynamic run-time requirements demand the flexibility to customize the model. However, as illustrated in Fig. 1a, we show that conventional static-model FL methods, represented FedAvg, cannot provide such customization. A naive solution is to train multiple models with different desired properties and keep them all on device. However, this leads to extra training and storage costs proportional to the number of models. Moreover, since it is not practical to keep all models simultaneously in run-time memory on a resource-limited device, it also introduces inference overhead to swap the models into and out of the run-time memory (Dosovitskiy & Djolonga, 2019).

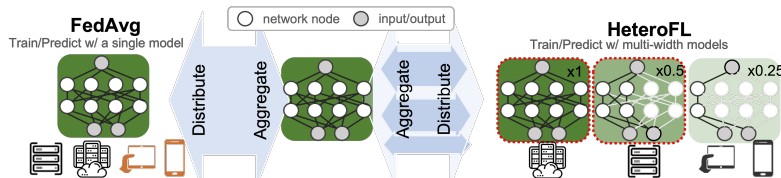

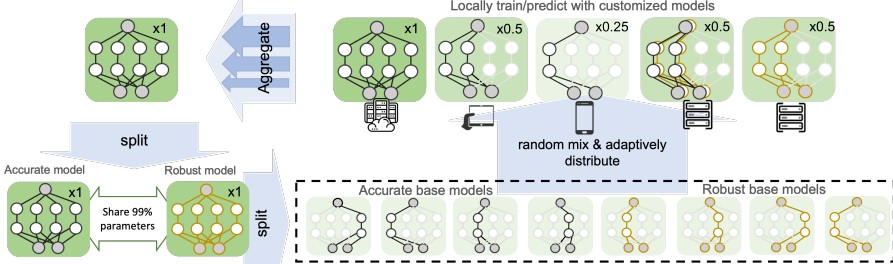

(a) Illustration of FedAvg (McMahan et al., 2017) with a device-incompatible model and a heterogenous-architecture variant (HeteroFL) (Diao et al., 2021) with under-trained wide models.

(b) The proposed Split-Mix framework provides in-situ customization of widths and adversarial robustness to address heterogeneity and dynamics, enabling efficient training and inference. In this example, we use a subnet with 1/4 channels (or widths) per layer as a base model for model-width customization. For simplicity, we denote it ×0.25 net, and a ×1 net can be split into 4 ×0.25 base models.

Figure 1: Comparison of traditional and proposed methods.

To effectively and efficiently train models for on-demand an in-situ customization, new challenges will be raised by the ubiquitous *heterogeneity* of federated learning participants. Fist, the participants can have *resource heterogeneity*: Different participants have different hardware resources available, such as memory, computing power, and network bandwidth (Ignatov et al., 2018). For example, in a learning task for face recognition, clients may use different types of devices (e.g., computers, tablets or smartphones) to participate in learning. To accommodate different hardware, one can turn to more resource-flexible architectures trained by distillation from ensemble (Lin et al., 2020), partial model averaging (Diao et al., 2021), or directly combining predictions (Shi et al., 2021). Specifically, (Diao et al., 2021) is the first heterogeneous-width solution (HeteroFL) allowing in-situ model-size switching. Nevertheless, it suffers from under-training in its large models due to local budget constraints as shown in Fig. 1a. The degradation could be worsened as facing *data heterogeneity*: The training datasets from participants are not independent and identically distributed (non-*i.i.d.*) (Li et al., 2020b; Fallah et al., 2020; Hong et al., 2021c; Zhu et al., 2021). When one device with a unique data distribution cannot afford training a large model, the global large model may not transfer to the unseen distribution (Pan & Yang, 2010). Thus, HeteroFL may not provide effective customization such that more parameters brings in higher accuracy and how to train an effectively customizable model still remains unknown.

To address the aforementioned challenges from heterogeneity and dynamics, we study a novel *Split-Mix* approach to enable FL on heterogeneous devices and achieve *in-situ model customization* for resource efficiency and robustness: The size and robustness of the resultant model can be efficiently customized at run-time. Specifically, we first **split** the complete knowledge in a large model into several small base sub-networks (shards) according to model widths and robustness levels. To complete the knowledge, we let the base models be fully trained on all clients. To provide customized models, we **mix** selected base models to construct the desired model size and robustness. We illustrate the main idea in Fig. 1b. Overall, our contributions can be summarized in three folds:

- Within the domain of heterogeneous federated learning, we are the first to study training a model with the capability of *in-situ customization* with heterogeneous local computation budgets, which cannot be resolved by existing methods yet as shown in Fig. 1a.

- To address the challenge, we propose a novel Split-Mix framework that aggregates knowledge from heterogeneous clients into a width- and robustness-adjustable model structure. Remarkably, due to fewer parameters and modular nature, our framework is not only efficient in federated communication and flexibly adaptable to various client budgets *during training*, but also efficient and flexible in storage, model loading and execution *during inference*.

- Empirically, we demonstrate that the performance of the proposed method is better than other FL baselines under heterogeneous budget constraints. Moreover, we show its effectiveness when facing the challenge of data heterogeneity.

## 2  RELATED WORK

**Heterogeneous Federated Learning.** As increasing concerns have been gained on data privacy leakage in machine learning (Dwork, 2008; Weiss & Archick, 2016; Wu et al., 2020; Hong et al., 2021b), federated learning (FL) protects data privacy by training the model locally on users' own devices without sharing data. In real-world applications, FL with budget-insufficient devices (e.g., mobile devices) has attracted a great amount of attention. For example, *FedDistill* (Lin et al., 2020) used logit averages to distill a network of the same size or several prototypes, which will be communicated with users. FedDistill made an assumption that the central server has access to a public dataset of the same learning task, which is impractical for many applications. He et al. (2020) introduced a distillation-based method after aggregating private representations from all participants. The method closely resembles centralized learning because all encoded samples are gathered, and however it is less efficient when local clients have large data dimensions or sample sizes. Importantly, the method may not transfer adversarial robustness knowledge through the intermediate representations due to the decoupling of the input and prediction. On the other hand, *HeteroFL* (Diao et al., 2021) avoids distillation, allowing participants to train different prototypes and sharing parameters among prototypes to reduce redundant parameters. However, HeteroFL also reduces the samples available for training each prototype, which leads to degraded performance. Considering the high cost of training robust models, Hong et al. (2021a) proposed an efficient way to transfer model robustness from budget-sufficient devices to insufficient ones. *FedEnsemble* (Shi et al., 2021) is technically related to the proposed approach, which uses ensemble of diverse models to accommodate non-*i.i.d.* heterogeneity. The authors showed that combining multiple base models trained simultaneously in FL can outperform a single base model in testing. A critical difference between the proposed approach and FedEnsemble comes from the challenging problem setting of constrained heterogeneous computation budgets. For the first time, we show that base models can be trained adaptively under budget constraints and used to customize efficient inference networks that can outperform a single model of the same width but trained in a heterogeneous-architecture way.

**Customizable Models.** To our best knowledge, this is the first paper discussing in-situ customization in federated learning and here we review similar concepts in central learning. First, customization of robustness and accuracy was discussed by Wang et al. (2020), where an adjustable-conditional layer together with decoupled batch-normalization were used. The conditional layer enables continuous trade-off but also brings in more parameters. In comparison, a simple weighted combination without additional parameters is used in our method and is very efficient in both communication and inference. In terms of model complexity, a series of work on dynamic neural networks were proposed to provide instant, adaptive and efficient trade-off between accuracy and latency (mainly related to model complexity) of neural networks at the inference stage. Typically, sub-path networks of different complexity (Liu & Deng, 2018) or sub-depth networks (Huang et al., 2018; Wu et al., 2019; Wang et al., 2018; Zhang et al., 2021) are trained together. However, due to the large memory footprint brought by a constant number of channels per layer, the memory footprint at inference is barely reduced. To address the challenge, slimmable neural network (SNN) (Yu et al., 2018) was proposed to train networks with adaptive layer widths. Distinct from SNN, we consider a more challenging scenario with distributed and non-sharable data and heterogeneous client capabilities.

## 3  PROBLEM SETTING

The goal of this work is to develop a heterogeneous federated learning (FL) strategy, that yields a global model, which can be efficiently customized for dynamic needs. Formally, FL minimizes the objective $\frac{1}{\sum_k |D_k|} \sum_{k=1}^{K} \sum_{(x,y) \in D_k} L(f(x; W), y)$, where $L$ is loss function (e.g., cross-entropy loss), $f$ is a model parameterized by $W$, and $\{D_k\}_{k=1}^{K}$ are the training datasets on $K$ participants with the image-label pairs $(x, y)$. Following the standard FL setting (McMahan et al., 2017), only

model parameters can be shared to protect privacy. We require efficient run-time customization on model $f$ for resource dynamics (Section 4.2) and robustness dynamics (Section 4.3). [1]

When training customizable/adjustable models, significant challenges arise from heterogeneity among clients. In this paper, we consider the following: **1) Heterogeneous computational budgets during training** of clients constrain the maximal complexity of local models. The complexity of deep neural networks can be defined in multiple dimensions, like depths or widths (Zagoruyko & Komodakis, 2017; Tan & Le, 2019). In this paper, we consider the width of hidden channels, because it can significantly reduce not only the number of model parameters but also the layer output cache. Thus, we assume clients have confined width capabilities $\{R_k \in (0, 1]\}_{k=1}^{K}$, defined by width ratios w.r.t. the original model, i.e., $\times R_k$ net as presented in Fig. 1b. Similar to FedAvg and HeteroFL, the same architecture is used by clients and therefore the model width can be tailored according to local budgets. In many applications, there are usually a significant number of devices with insufficient computational budgets (Enge, 2021). For example, we may assume exponentially distributed budgets in uniformly divided client groups: $R_k = (1/2)^{\lceil 4k/K \rceil}$, i.e., the first group with $1/4$ clients is capable for a 1-width and the rest are for $0.5, 0.25, 0.125$-widths respectively. The budget distribution simulate a real scenario where most federated mobile phones prefer a low-energy solution with smaller models for training probably in the background and maintain more resources for major tasks. Other budget distributions, e.g., log-normal distributions, are discussed in Appendix A.5. **2) Heterogeneous data distributions**, e.g., non-*i.i.d.* features, $D_k \sim \mathcal{D}_i$ for ordered domain $i$, induces additional challenges with a skewed budget distribution, since training a single model on multiple domains (Li et al., 2020b) or models in a single domain (due to budget constraints) (Hong et al., 2021c; Dong et al., 2021) are known to be suboptimal on transfer performance.

In summary, training various model sizes has challenges from 1) resource heterogeneity, where disparity in computation budgets will significantly affect the training of large models because they can only be trained on scarce budget-sufficient clients; 2) data heterogeneity, where the under-trained large models may perform poorly on unseen in-domain or out-of-domain samples.

## 4 METHOD

To provide efficient *in-situ* customization, we introduce a simple yet powerful principle, Split-Mix: *shatter complete knowledge into smaller pieces, and customize by flexible formations of pieces.* Based on the principle, we propose customizations of model size and robustness in the following.

### 4.1 CASE STUDY: CUSTOMIZABLE NETWORKS FROM BUDGET-CONSTRAINED FL

For motivation, we set up a standard non-*i.i.d.* FL benchmark (Li et al., 2020b) using DomainNet dataset (refer to experimental details in Section 5) and study how the budget constraint impedes effective training of customizable models.

**Case 1: FL without budget constraint.** First, we individually train networks of different widths by FedAvg. We see that the slimmer networks converge slower and are less generalizable, even though they are trained on all clients (solid lines in Fig. 2). The results justified the motivation of training wider networks than slimmer ones.

**Case 2: FL with budget constraint.** Following the above budget constraint, i.e., $R_k = (1/2)^{\lceil 4k/K \rceil}$, we deploy HeteroFL to train budget-compatible prototype models locally. Since HeteroFL was not designed for model customization, data are not fully used: each model prototype is only trained by $1/4$ of clients, when clients of the $R_k = 1$ budget can actually afford all slimmer models. Therefore, we extend HeteroFL to a

Figure 2: Convergence of different-width models on DomainNet.

*slimmable* version (SHeteroFL) by training all affordable prototypes locally. As shown in Fig. 2, wider models (e.g., $\times 1$ net) converge to validation accuracy lower than not only the FedAvg counterparts,

---

[1]Customization of other properties can also be applied within our framework. We consider model size and robustness given their practical importance.

but also the slimmer $\times 0.25$ net, showing that the widest model may not be a good candidate model. Therefore, switching models to wider configurations lowers efficiency but does not improve accuracy, which is *not* a valid customization.

From the perspective of data allocation, it is not surprising that SHeteroFL exhibits a non-monotonous relation between the model width and accuracy. For $\times 1$ nets, only $1/4$ clients and data are accessible for training. In comparison, $3/4$ data are accessible by $\times 0.25$ nets and the more data empowers $\times 0.25$ nets better generalization ability than $\times 1$ nets.

## 4.2    CUSTOMIZE MODEL SIZE

Motivated by the above observations, we propose to increase accessible training data by splitting wide networks into universally-budget-compatible sub-networks and re-mix afterward. The overall FL algorithm is summarized in Algorithm 1.

**More accessible data by splitting wide networks.** Since a wide network cannot fit into budget-insufficient clients, we split it into budget-compatible sub-networks by channels (width) while maintaining the the total width. In terms of memory limitations, each sub-network can be painlessly and individually trained in all clients. However, sequentially training multiple slim base networks could be much slower than training a single integrated one and increases blocking time on communication. Noticing that all base models can be evaluated independently, we instead efficiently train $\times r$ base models in parallel (see Algorithm 3). Despite the benefit, a client's budget constraint ($R$) will limit the number of paralleled base models within $\lfloor R/r \rfloor$, as wider channels result in larger intermediate layer cache (activations), and excludes the rest base models from the client. Fortunately, we can select different sets of base models for a budget-limited client per round, which is inspired by FedEnsemble (Shi et al., 2021) and is presented in Algorithm 2. Hence, all base models can be ever trained on the client for multiple communication rounds, though not continuously every round. Note that since the federated training processes of base models are independent without interference, the training could be as stable as FedAvg with partial participants. In addition, the combination of slimmer base models is flexible and can conform a variety of client budgets.

**Boost accuracy by mixture of subnet experts.** To craft a wide model, we combine the outputs of multiple $\times r$ base models until the size of the ensemble reaches the same as the number of channels, e.g., $\lfloor R/r \rfloor$ bases for an $\times R$ net. We randomly initialize base models independently such that the diverse bases could extract different features from the same image (Allen-Zhu et al., 2020). Therefore, the ensemble can predict based on a variety of features, resembling an integrated wide network. We follow the common practice, Kaiming's method (He et al., 2015), for initializing base networks with ReLU layers. As Kaiming's method is width-dependent, we parameterize the initialization based on the width of $\times 1$ net instead of the $\times r$ one, which leads to smaller initial convolutional weights. In Appendix A.3, intensive ablation studies show that the rescaled initialization is critical for improving test accuracy of wider networks.

---

**Algorithm 1** Federated Split-Mix Learning

**Input**: Client datasets $\{D_k\}_{k=1}^K$, the number of total communication rounds $T$, $M = \lfloor 1/r \rfloor$ randomly initialized base $\times r$ nets parameterized by $\{w_{i,r}\}_{i=1}^M$, client budgets $\{R_k\}_{k=1}^K$, the number of local epochs $E$, learning rates $\{\eta_t\}_{t=1}^T$

1: Initialize $\{w_{i,r}^0\}_{i=1}^M$, model indexes $P = \text{Shuffle}([1, \cdots, M])$ and current index $p = 1$
2: **for** round $t \in \{1, \cdots, T\}$ **do**
3:      Initialize $W^t \leftarrow \{w_{i,r}^t \leftarrow 0\}_{i=1}^M$ and aggregation weights $c_i = 0$ for $i \in \{1, \ldots, M\}$
4:      **for** $k \in \{1, \cdots, K\}$ **do**
5:          Sample base models $W_k^{t-1}, p \leftarrow \text{SampleBaseModels}(P, p, \lfloor R_k/r \rfloor, W^{t-1})$
6:          Send $W_k^{t-1}$ to client $k$ and train $\hat{W}_k^t \leftarrow \text{LocalTrain}(W_k^{t-1}, D_k, E, \eta_t)$
7:          Aggregate $\hat{w}_{i,r}^{t,k} \in \hat{W}_k^t$ to the server: $w_{i,r}^t \leftarrow w_{i,r}^t + \hat{w}_{i,r}^{t,k}|D_k|, c_i \leftarrow c_i + |D_k|$
8:      Server update $W^t \leftarrow \{w_{i,r}^t \leftarrow w_{i,r}^t/c_i\}_{i=1}^M$
9: (Optional) Sort $W^T = \{w_{i,r}^T\}_{i=1}^M$ by the descending order of the validation accuracy of $w_{i,r}^T$
10: **Output** the customizable model $W^T = \{w_{i,r}^T\}_{i=1}^M$
11: **Customize** an $R$-width model by $F(x; W^T) = \frac{1}{K_R} \sum_{i=1}^{K_R} f(x; w_{i,r}^T)$ where $K_R = \lfloor R/r \rfloor$

---

| **Algorithm 2** SampleBaseModels($P, p, n, W$) | **Algorithm 3** LocalTrain($W_k, D_k, E, \eta$) |
|---|---|
| 1: **if** $p > \|P\|$ **then** Shuffle $P$ and $p \leftarrow 1$ | 1: Initialize models $\hat{W}_k$ by $W_k$ |
| 2: Initialize $W = \{w_{P[p],r}\}$ | 2: **for** $e \in 1, \cdots, E$ **do** |
| 3: **if** n>1 **then** | 3:     **for** batch data $(x, y)$ in $D_k$ **do** |
| 4:     Uniformly sample $n - 1$ values into $S$ from $P \backslash \{P[p]\}$ without replacement | 4:         **for** $\hat{w}_{i,r} \in \hat{W}_k$ in parallel **do** |
| 5:     $W \leftarrow W \cup \{w_{i,r}, \forall i \in S\}$ | 5:            $\hat{w}_{i,r} \leftarrow \hat{w}_{i,r} - \eta \frac{\partial L(f(x; \hat{w}_{i,r}), y)}{\partial \hat{w}_{i,r}}$ |
| 6: **Return** $W, p + 1$ | 6: **Return** $\hat{W}_k$ |

## 4.3 EXTENSION TO ADVERSARIAL ROBUSTNESS CUSTOMIZATION

In this section, we extend the customization from one dimension to two dimensions, by jointly customizing model size and model robustness under adversarial attacks (Goodfellow et al., 2014). Model robustness has gained increasing interest (Hendrycks & Dietterich, 2018; Wang et al., 2021), especially in high-stakes federated learning applications (Hong et al., 2021a). Adversarial training (AT) (Madry et al., 2018) is arguably the most popular and effective defense strategy against adversarial attacks. Specifically, it uses on-the-fly adversarial samples as augmentation to improve robustness. Formally, AT minimizes the following augmented loss:

$$L(f) = (1 - \lambda_n) L_{CE}(f(x), y) + \lambda_n \max_{\|\delta\|_\infty \leq \epsilon} L_{CE}(f(x + \delta), y), \tag{1}$$

where $\delta$ is a subtle $\epsilon$-constrained adversarial noise and transfers a clean sample $x$ into an *adversarial sample* $x + \delta$. In Eq. (1), $L_{CE}$ is the cross-entropy loss the hyper-parameter and $\lambda_n$ trades off accuracy (the 1st term) and robustness (the 2nd term). When $\lambda_n = 0$ or 1, the optimization yields a *standard-training* (ST) model or an AT model, respectively. Since an AT model is commonly less accurate in predicting standard images (Tsipras et al., 2019), there is usually no such a sweet point of $\lambda_n$ simultaneously maximizing robustness and accuracy, and one typically needs to carefully gauge the trade-off according to the demand of robustness in specific application context.

**Splitting and sharing parameters.** Since standard performance and adversarial robustness are irreconcilable, we can directly use two separated ST and AT models to maximally capture the each property. But do we really need two totally separated models? Intuitively, the two models share some common knowledge, given that an adversarial image share a large part of common features with its original version. As introduced by (Xie & Yuille, 2019), sharing all parameters except the batch-normalization (BN) layers can maximize robustness and accuracy by

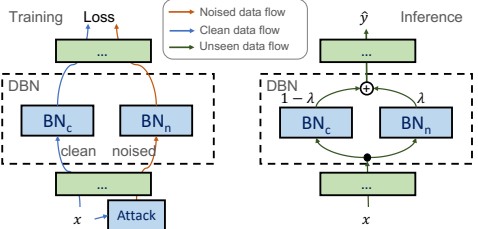

Figure 3: Illustration of dual batch-normalization (DBN) in training and inference. The $\text{BN}_c$ and $\text{BN}_n$ are for clean and noised samples.

expertised BNs, respectively. Accordingly, we propose to split BN layers (instead of the whole model) into two sub-components: one for standard performance and the other for robustness. At training time, the first loss of Eq. (1) is computed with clean BN ($\text{BN}_c$) merely and the second adversarial loss is computed with noised BN ($\text{BN}_n$). The FL local training is elaborated in Algorithm 4. As the two BNs are decoupled, there is no more trade-off in loss Eq. (1) and thereby we choose $\lambda_n = 0.5$ to balance their effects on parameter updates.

**Customizable layer-wise mixing.** After training, the problem is how to mix the two models (with different BNs) for prediction. A straightforward solution is averaging their outputs. However, forwarding memory footprint will be doubled in this way, as the two models have to be both executed separately. To avoid the doubled intermediate outputs, a gate function can be used to adaptively choose BNs like (Liu et al., 2020). Inspired by the method, we further propose a simple parameter-free method by weighted-averaging outputs of each BN layer (see Fig. 3):

$$\text{DBN}(x) = (1 - \lambda)\text{BN}_c(x) + \lambda \text{BN}_n(x), \tag{2}$$

given a BN-layer input $x$. We note that the averaging strategy is entirely training-free and does not use extra parameters, and the customization weight $\lambda$ is intuitive for trade-offs between SA and RA.

Lastly, it is remarkable that the DBN structure is rather lightweight in terms of model complexity. As investigated in (Yu et al., 2018) (Table 2), the parameters of BN is no more than $1\%$ in popular deep

architectures, e.g., ResNet or MobileNet. Therefore, we can plug DBN into base models in place of BN, replace Algorithm 3 with Algorithm 4, and jointly customize robustness and model widths.

## 5 EMPIRICAL STUDIES

We design experiments to compare the proposed method against FL classification benchmarks. For **class non-*i.i.d*** configuration, we use CIFAR10 dataset (Krizhevsky, 2009) with preactivated ResNet (PreResNet18) (He et al., 2016). CIFAR10 contains over $50,000$ $32 \times 32$ images of 10 classes. The CIFAR10 data are uniformly split into 100 clients and distribute 3 classes per client. For **(feature) non-*i.i.d.*** configuration, we use Digits with a CNN defined and DomainNet datasets (Li et al., 2020b) with AlexNet extended with BN layers after each convolutional or linear layer (Li et al., 2020b). The first dataset is a subset (30%) of Digits, a benchmark for domain adaption (Peng et al., 2019b). Digits has $28 \times 28$ images and serves as a commonly used benchmark for FL (Caldas et al., 2019; McMahan et al., 2017; Li et al., 2020a). The dataset includes 5 different domains: MNIST (Lecun et al., 1998), SVHN (Netzer et al., 2011), USPS (Hull, 1994), SynthDigits (Ganin & Lempitsky, 2015), and MNIST-M (Ganin & Lempitsky, 2015). The second dataset is DomainNet (Peng et al., 2019a) processed by (Li et al., 2020b), which contains 6 distinct domains of large-size $256 \times 256$ real-world images: Clipart, Infograph, Painting, Quickdraw, Real, Sketch. Each domain of Digits (or DomainNet) are split into 10 (or 5) clients, and therefore 50 (or 30) clients in total. We defer other details such as hyper-parameters to Appendix A, and focus on discussing the results.

### 5.1 CUSTOMIZE MODEL SIZES

In this section, we evaluate the proposed Split-Mix on tasks of customizing model sizes through adjusting model widths. Recall that in Section 3, we assume a specific heterogeneous training budget to facilitate our discussion, such that one client can only train models within a maximal width, and the resource distribution is imbalance among clients: $R_k = (1/2)^{\lceil 4k/K \rceil}$. Following the common FL setting, we do not consider algorithms that use public data or representation sharing in our baselines.

**Baselines**. As an ideal upper bound but a memory-incompatible baseline, we (re-)train networks from scratch by FedAvg to obtain individual models with different widths. The state-of-the-art heterogeneous-architecture FL method is *HeteroFL* (Diao et al., 2021), which trains different slim models in different clients. For a fair comparison, we extend HeteroFL with bounded-slimmable training in clients who can afford the larger models, named *SHeteroFL*. For example, if a client in HeteroFL can afford $\times 0.5$ net, then the client meanwhile trains $\times 0.25$ net and other smaller subnets in slimmable training manner (Yu et al., 2018) by SHeteroFL.

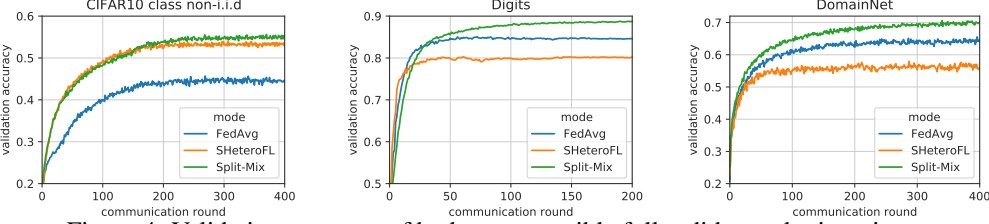

Figure 4: Validation accuracy of budget-compatible full-wdith nets by iterations.

**Convergence.** In Fig. 4, we compare the convergence curves of full-width models on three datasets. For FedAvg, the budget-compatible width is $\times 0.125$. Both for SHeteroFL and Split-Mix, the full-width is $\times 1$. Compared to baselines, the proposed Split-Mix converges faster, largely due to the splitting strategy. Specifically, because all base models are independently trained, the convergence of each base model mainly depends on how frequently they are trained. When enough clients participate FL, the training frequency of the $\times 1$ net is more than one, as all bases will be selected at least once in a communication round.

**Performance.** In Table 1, we compare the test accuracy of different model widths. We measure the latency in terms of MACs (number of multiplication-and-addition operations) and model size in parameter numbers. With the same width, our method outperforms SHeteroFL, using much fewer parameters and thus conducts inference in much lower latency. Remarkably, compared to the best model by SHeteroFL, e.g., $81.8\%$ $\times 1$ net in CIFAR10, Split-Mix uses only 1.8% parameters and

Table 1: Test results of customizing model width. MACs and the number of parameters are counted at inference time. Grey texts indicate that the training cannot conform the predefined budget constraint. The 'M' after metric values means $\times 10^6$.

| width | Individual FedAvg | | | SHeteroFL | | | Split-Mix (ours) | | |
|---|---|---|---|---|---|---|---|---|---|
| | Acc | MACs | #Params | Acc | MACs | #Params | Acc | MACs | #Params |
| | CIFAR10 class non-*i.i.d* FL | | | | | | | | |
| ×0.125 | 43.4% | 0.9M | 0.2M | **49.1%** | 0.9M | 0.2M | 48.0% | 0.9M | 0.2M |
| ×0.25 | 45.7% | 3.5M | 0.7M | **51.4%** | 3.5M | 0.7M | 51.1% | **1.8M** | **0.4M** |
| ×0.5 | 50.3% | 14.0M | 2.8M | 51.5% | 14.0M | 2.8M | **52.1%** | **3.6M** | **0.7M** |
| ×1 | 53.3% | 55.7M | 11.2M | 49.9% | 55.7M | 11.2M | **52.7%** | **7.2M** | **1.4M** |
| | Digits feature non-*i.i.d* FL | | | | | | | | |
| ×0.125 | 86.1% | 0.1M | 0.2M | **86.8%** | 0.1M | 0.2M | 84.6% | 0.1M | 0.2M |
| ×0.25 | 87.3% | 0.4M | 0.9M | **87.9%** | 0.4M | 0.9M | 87.5% | **0.2M** | **0.4M** |
| ×0.5 | 88.7% | 1.3M | 3.6M | 86.9% | 1.3M | 3.6M | **89.0%** | **0.5M** | **0.9M** |
| ×1 | 89.6% | 4.8M | 14.2M | 81.3% | 4.8M | 14.2M | **89.8%** | **0.9M** | **1.8M** |
| | DomainNet feature non-*i.i.d* FL | | | | | | | | |
| ×0.125 | 67.2% | 2.5M | 0.9M | 66.9% | 2.5M | 0.9M | **68.4%** | 2.5M | 0.9M |
| ×0.25 | 69.8% | 7.5M | 3.6M | 67.8% | 7.5M | 3.6M | **71.9%** | **5.0M** | **1.8M** |
| ×0.5 | 72.6% | 25.5M | 14.3M | 66.9% | 25.5M | 14.3M | **73.0%** | **9.9M** | **3.6M** |
| ×1 | 72.9% | 92.5M | 57.1M | 58.7% | 92.5M | 57.1M | **74.2%** | **19.8M** | **7.2M** |

1.6% MACs (×0.125 net) to reach a similar level of test accuracy. We notice that SHeteroFL has a much lower test accuracy with ×0.125 net on the CIFAR10 dataset. By investigating the loss curves, we find that the inference between parameter-shared different prototypes results in unstable convergence of the ×0.125 net. Attributed to the independent splitting, our method is more stable on convergence with different widths. Remarkably, Split-Mix only requires 12.7% parameters and 19.8% MACs (by ×1 ensemble) to achieve the comparable accuracy as the ×1 individual network on Digits. Results on CIFAR10 and DomainNet show a potential limitation of our method. The accuracy of Split-Mix is not comparable with the wider individual models due to the locally limited complexity. However, the limitation is more from the problem setting itself, and will not undermine our advantage in budget-limited FL, since wide individual models cannot be trained in this case.

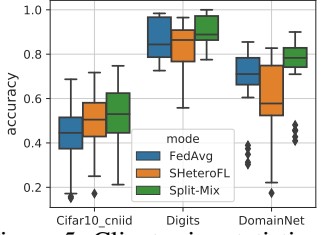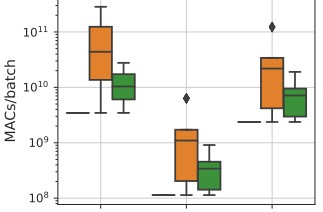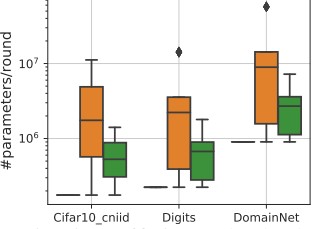

Figure 5: Client-wise statistics of test accuracy, training and communication efficiency by budget constraints. The MACs quantify the complexity of one batch optimization in a client, and the number of parameters per round round are the ones uploaded to (or downloaded from) a server. Test accuracy is by the full-width networks. The results of FedAvg are from budget-compatible ×0.125 nets.

**Client-wise evaluation.** In addition to comparisons of inference on average, we also demonstrate the statistics of test accuracy and the training and communication efficiency in Fig. 5. Conforming the budget constraints, our method outputs more accurate full-width models, transfer fewer parameters per round and execute fewer multiplication-and-add operations for gradient descent than SHeteroFL, either in terms of average or variance. Because only the individual ×0.125 net can fit into the budget constraint, FedAvg requires the least MACs and parameters, which however significantly sacrifices the final accuracy.

**Domain-wise evaluation.** To understand why Split-Mix outperforms SHeteroFL, we investigate the total percentage of parameters that can be trained in each domain, in Fig. 6 (Left). We count the total parameters that were ever trained in clients of a domain during the learning. Thanks to the base-model sampling strategy (i.e., Algorithm 2), Split-Mix allows all base models, rather than a subset, to be trained on all clients. On the other hand, varying client budgets greatly impacted SHeteroFL by limiting the width of models trained in budget-insufficient clients, e.g., in the clipart, infograph and painting domains. Hence, SHeteroFL leaves a large amount of parameters under-trained and suffers

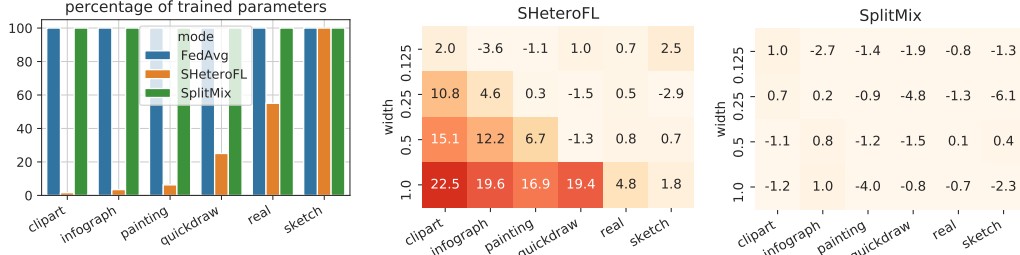

Figure 6: Per domain in DomainNet, the total percentage of parameters that are locally trained (the left figure) and the accuracy (%) drops compared to FedAvg individual models (right two figures). from larger accuracy losses in the three domains and wider models. In comparison, Split-Mix not only has less accuracy drop but is also more stable in all domains.

## 5.2 CUSTOMIZE ROBUSTNESS

**Training and evaluation.** For local AT, we use an $n$-step projected gradient descent (PGD) attack (Madry et al., 2018) with a constant noise magnitude $\epsilon$. Following (Madry et al., 2018), we set $(\epsilon, n) = (8/255, 7)$, and attack inner-loop step size $2/255$, for training, validation, and test. For simplicity, we temporarily *relax the budget constraint $R_k$ and let the base model be $\times 1$ net.* For comparison, we extend FedAvg with AT which yields individual models optimized with different trade-off variables, i.e., $\lambda_n \in \{0, 0.1, 0.2, 0.3, 0.5, 1\}$. Also, we extend FedAvg with state-of-the-art in-situ trade-off method, OAT (Wang et al., 2020), as a baseline. Split-Mix+DAT is an extension of Split-Mix by the proposed DBN-based AT in Algorithm 4. We evaluate and contrast models in two metrics: *standard accuracy (SA)* on the clean test samples and *robust accuracy (RA)* on adversarial images generated from the clean test set by the PGD attack. Both metric values are averaged by users. We evaluate the trade-off effectiveness by comparing the RA-SA curves, which is better approaching the right-upper corner. To plot the curves for FedAvg+OAT and Split-Mix+DAT, we vary their condition variable $\lambda$ in $\{0, 0.2, 0.5, 0.8, 1\}$.

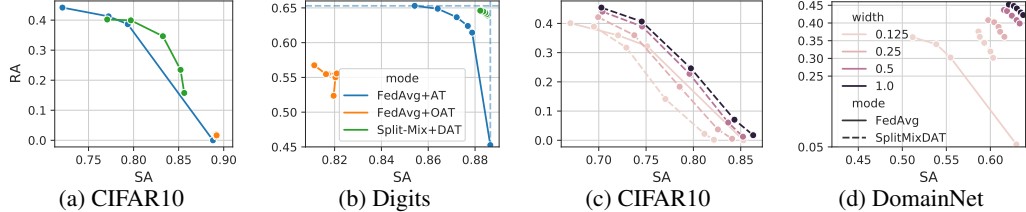

| (a) CIFAR10 | (b) Digits | (c) CIFAR10 | (d) DomainNet |

Figure 7: Trade-off between robust accuracy (RA) and standard accuracy (SA) with full width (a,b) and customizable widths (c,d).

**Trade-off curves** are presented in Fig. 7 (a) and (b). Since the naive extension of OAT with FedAvg adopts heterogeneous objectives and over-parameterization of conditional layers which suffers from convergence issues, its adversarial training does not converge and RA is incredibly poor in some cases. As a result, the trade-off curve is not smooth. Instead, the proposed Split-Mix+DAT method has a smoother trade-off curve without heavy conditional training or over-parameterization. By training in one pass, Split-Mix+DAT even outperforms and is more efficient than the FedAvg+AT baselines.

**Joint customization of width and robustness under budget constraints.** Now we consider the width customization and the training budgets as Section 5.1. Due to the constraint, FedAvg can only train $\times 0.125$ net. We omit OAT for its unstable convergence and use SplitMixDAT as a short name of Split-Mix+DAT. In Fig. 7 (c) and (d), the trade-off curves with different widths are depicted. As the width increases, both RA and SA of SplitMixDAT are improved, when they are smoothly traded off. The results demonstrate the flexibility and the modular nature of our method.

## 6 CONCLUSION

In this paper, we proposed a novel federated learning approach for in-situ and on-demand customization to address challenges arise from resource heterogeneity and inference dynamics. We proposed a Split-Mix strategy that efficiently transfer clients' knowledge to collaboratively learn a customizable model. Extensive experiments demonstrate the effectiveness of the principle in adjusting model widths and robustness when much fewer parameters are used compared to baselines.

ACKNOWLEDGMENTS

This material is based in part upon work supported by the National Institute of Aging 1RF1AG072449, Office of Naval Research N00014-20-1-2382, National Science Foundation under Grant IIS-1749940. Z.W. is supported by the U.S. Army Research Laboratory Cooperative Research Agreement W911NF17-2-0196 (IOBT REIGN).

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

# A  EXPERIMENTS

In this section, we provide more details about our experiments and additional evaluation results.

## A.1  PARALLEL IMPLEMENTATION OF SPLIT-MIX AND CONVERGENCE

In this section, we elaborate on the implementation and efficiency of Split-Mix. In Fig. 8, we conceptually compare the training by three methods, when two clients capable of training $\times 1$ and $\times 0.5$ are considered. FedAvg can train the $\times 1$ net on the $\times 1$-capable client but not on the $\times 0.5$-capable client. Through parameter sharing among different model widths, SHeteroFL can train multiple widths in a sequential manner and can fit into $\times 0.5$-capable clients. Unlike SHeteroFL, Split-Mix trains four base models in one parallel pass and therefore is the most efficient and flexible method. Remarkably, in the worst case, training $\times 1$ net of Split-Mix is as efficient as FedAvg and more efficient than SHeteroFL, if all the base weights, $w_{0,r}^l, \ldots, w_{3,r}^l \in \mathbb{R}^{r \times r}$ (where $r = 0.25$ here), are embedded into a $4r \times 4r$ weight matrix[2]. When base models are embedded into a full net, non-trainable parameters can be masked out (grey areas) to avoid interference between base models.

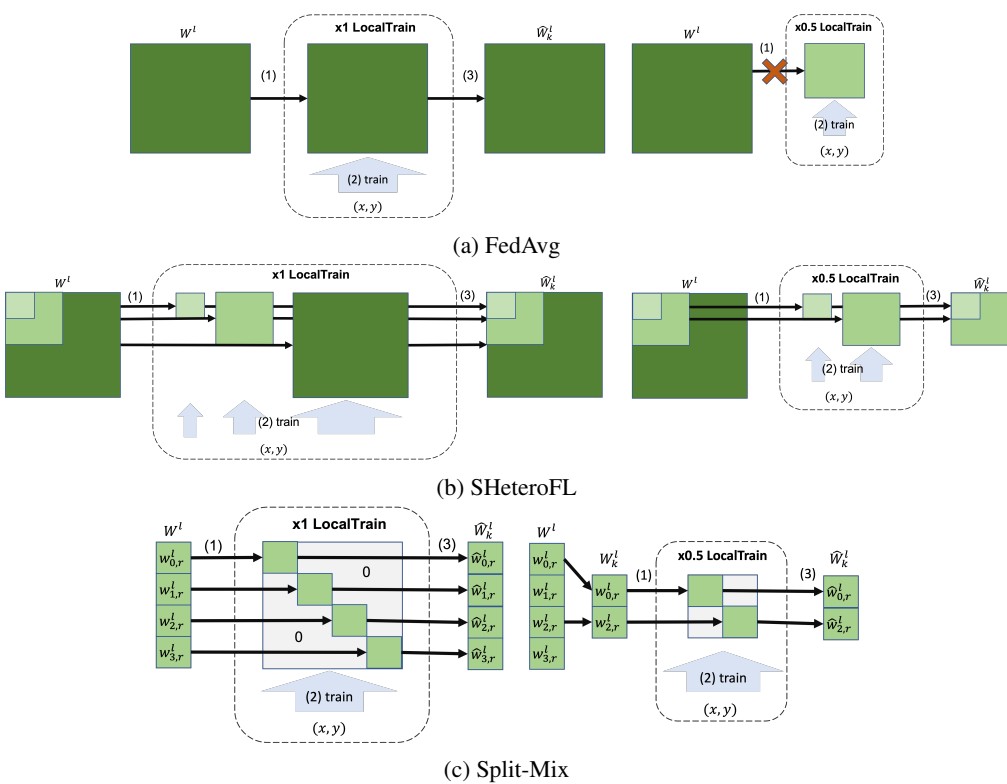

(a) FedAvg

(b) SHeteroFL

(c) Split-Mix

Figure 8: Illustration of training weight matrices on a $\times 1$-net-capable or $\times 0.5$-net-capable client. (1) Download the global weight matrix $W^l$ of layer $l$ or a selected subset $W_k^l$. (2) Train weights on a batch data $(x, y)$. (3) Upload trained weight matrix $\hat{W}_k^l$.

With the aforementioned implementation, we compare the convergence versus the wall-clock time in Fig. 9. We implement all algorithms in `PyTorch 1.4.1` run on a single NVIDIA RTX A5000 GPU and a 104-thread CPU. Fig. 9 shows the elapsed computation time from the initialization to a maximal number of iterations. The maximal number of iterations is set to be the same for all methods, such that it is easier to compare the stopping time. In Fig. 9, Split-Mix is much more efficient than the SheteroFL. Note that FedAvg can only train the $\times 0.125$ net which includes much fewer parameters, For this reason, Split-Mix is slightly slower than FedAvg, but the degradation of efficiency trades in better accuracy than FedAvg.

---

[2]For the simplicity of notations, we assume the width of layer $l$ is $r$, though the width could vary by layer in general.

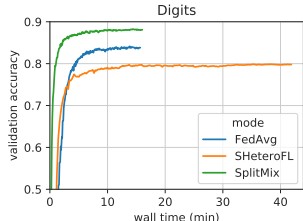

Figure 9: Validation accuracy of the budget-compatibly-widest nets by wall-clock time. All algorithms are run with the same number of iterations (200).

## A.2 EXPERIMENTAL CONFIGURATIONS

In Algorithm 4, we elaborate the local training of DBN. Specifically, each BN is trained independent with different input samples. The maximization problem in Algorithm 4 can be solved by an $n$-step projected gradient descent, and is commonly known as *PGD attack* (Madry et al., 2018).

---

**Algorithm 4** LocalTrain($W_k, D_k, E, \eta$) with DBN and adversarial training

---

1: Initialize models $\hat{W}_k$ by $W_k$
2: **for** $e \in \{1, \cdots, E\}$ **do**
3:      **for** mini-batch $B = \{(x, y)\}$ in $D_k$ **do**
4:          **for** $\hat{w}_{i,r} \in \hat{W}_k$ in parallel **do**
5:              Set $f$ to use clean BN
6:              $L \leftarrow \frac{1}{|B|} \sum_{(x,y) \in B} L_{CE}(f(x; \hat{w}_{i,r}), y)$
7:              Set $f$ to use noise BN
8:              $\tilde{B} = \emptyset$
9:              **for** $x \in B$ **do**
10:                  Perturb $\tilde{x} = x + \delta$ with $\delta \leftarrow \arg\max_{\|\delta\|_\infty \leq \epsilon} L_{CE}(f(x + \delta; \hat{w}_{i,r})), y)$
11:                  $\tilde{B} \leftarrow \tilde{B} \cup \{(\tilde{x}, y)\}$
12:              $L \leftarrow \frac{1}{2} \left\{ L + \frac{1}{|\tilde{B}|} \sum_{(\tilde{x},y) \in \tilde{B}} [L_{CE}(f(\tilde{x}; \hat{w}_{i,r}), y)] \right\}$
13:              $\hat{w}_{i,r} \leftarrow \hat{w}_{i,r} - \eta \frac{\partial L}{\partial \hat{w}_{i,r}}$
14: Return $\hat{W}_k$

---

**Data**. Both CIFAR10 and Digits are 10-way classification tasks. We follow the non-*i.i.d* benchmark of Li et al. (2020b) to extract 10 classes from DomainNet, which is publicly available in FedBN codes. To illustrate the multiple-domain datasets, we sample several images from Digits and DomainNet datasets in Fig. 10.

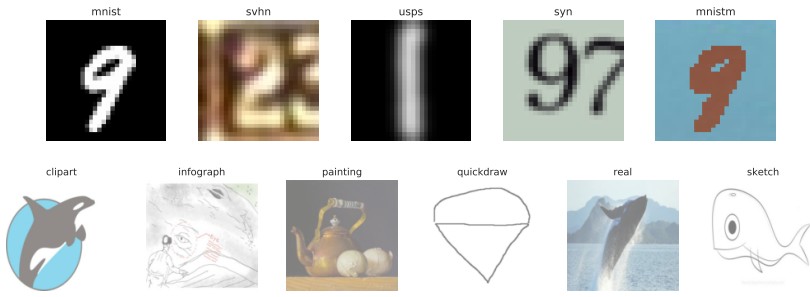

Figure 10: Sample images from multiple domain datasets.

**Hyper-parameters.** In general, for local optimization we use stochastic gradient descent (SGD) with 0.9 momentum and $5 \times 10^{-4}$ weight decay. Dataset specific settings are stated as follows. CIFAR10: Following HeteroFL (Diao et al., 2021), we train with 5 local epochs and 400 global

Table 2: Network architecture for Digits dataset.

| Layer | Details |
|---|---|
| **feature extractor** | |
| conv1 | Conv2D(64, kernel size=5, stride=1, padding=2) |
| bn1 | DBN2D, RELU, MaxPool2D(kernel size=2, stride=2) |
| conv2 | Conv2D(64, kernel size=5, stride=1, padding=2) |
| bn2 | DBN2D, ReLU, MaxPool2D(kernel size=2, stride=2) |
| conv3 | Conv2D(128, kernel size=5, stride=1, padding=2) |
| bn3 | DBN2D, ReLU |
| **classifier** | |
| fc1 | FC(2048) |
| bn4 | DBN2D, ReLU |
| fc2 | FC(512) |
| bn5 | DBN1D, ReLU |
| fc3 | FC(10) |

communication rounds. Globally, we initialize the learning rate as 0.01 and adjust the learning rate at 150, 250 communication rounds with a scale rate of 0.1. Locally, we use a larger batch size of 128, to speed up the training in simulation. Digits: We use a cosine annealing learning rate decaying from 0.1 to 0 across 400 global communication rounds. SGD is executed with one epoch for each local client. DomainNet: We use a constant learning rate 0.01 and run 400 communication rounds in total. Similar to Digits, SGD is executed with one epoch for each local client.

Table 3: Network architecture for DomainNet dataset.

| Layer | Details |
|---|---|
| **feature extractor** | |
| conv1 | Conv2D(64, kernel size=11, stride=4, padding=2) |
| bn1 | DBN2D, ReLU, MaxPool2d(kernel size=3, stride=2) |
| conv2 | Conv2D(192, kernel size=5, stride=1, padding=2) |
| bn2 | DBN2D, ReLU, MaxPool2d(kernel size=3, stride=2) |
| conv3 | Conv2D(384, kernel size=3, stride=1, padding=1) |
| bn3 | DBN2D, ReLU |
| conv4 | Conv2D(256, kernel size=3, stride=1, padding=1) |
| bn4 | DBN2D, ReLU |
| conv5 | Conv2D(256, kernel size=3, stride=1, padding=1) |
| bn5 | DBN2D, ReLU, MaxPool2d(kernel size=3, stride=2) |
| avgpool | AdaptiveAvgPool2d(6, 6) |
| **classifier** | |
| fc1 | FC(4096) |
| bn6 | DBN1D, ReLU |
| fc2 | FC(4096) |
| bn7 | DBN1D, ReLU |
| fc3 | FC(10) |

**Network architectures**. Architectures of modified AlexNet (for DomaiNet) and CNN (for Digits) can be found in (Li et al., 2020b) and public codes [3]. For reader's reference, we provide the layer details in in Tables 2 and 3. For the convolutional layer (Conv2D or Conv1D), the first argument is the number channel. For a fully connected layer (FC), we list the number of hidden units as the first argument. The implementation of the preactivated ResNet can be found in the repository of HeteroFL (Diao et al., 2021).

**Batch-normalization for customizable model sizes**. As pointed out by Diao et al. (2021), HeteroFL relies on mini-batch batch-normalization to stabilize the training with multiple model widths and

---

[3]https://github.com/med-air/FedBN

compute the statistics afterwards. Another advantage of such strategy is on federation of multi-domain clients. Li et al. (2020b) showed that using local batch-normalization helps model personalization for non-*i.i.d* local features. As min-batch BN statistics simulate the local BN idea, it should enjoy the similar benefit. To eliminate the potential biases in personalization or convergence caused by different estimation methods of BN statistic, we let *all* compared algorithms using the same mini-batch strategy. To reveal the effect of different BN statistic solutions for Split-Mix, we provide a detailed ablation study regarding the estimation of BN statistics in Appendix A.3.

**Loss function for class non-*i.i.d* FL**. As some classes are missing locally in non-*i.i.d* setting, class-specific parameters in the classifier head may be updated without proper supervision and results in random updates. To mitigate the effect of missing classes locally, we use the same masked cross-entropy loss as introduced by HeteroFL, where absent classes are masked out.

## A.3 ABLATION STUDY OF NETWORK SCALING AND BN STATISTICS

In this section, we evaluate how the network rescaling and BN statistics affect the performance. For rescaling, we consider the parameter initialization (*rescale init*) and layer outputs (*rescale layer*). For BN statistics, we consider four options. The *batch average* one will estimate the statistics by one batch of data. The *post average* one will use the batch average strategy during training but re-estimate the statistics using client data afterward, which was adopted by HeteroFL. To gain better BN statistics, we run the model on a training set for 20 epochs. The *tracked* one will track statistics during training. The *locally tracked* one will also track statistics but the statistics will not be shared with the server for averaging, which can benefit clients' privacy and personalization (Li et al., 2020b).

We report full ablation results in Table 4. **1)** First we compare the use of BNs without in-training tracking. The post-average BN performs best compared to other BN choices. With similar performance, the batch average BN does not need multiple rounds of evaluation of BN statistics, which is more efficient for inference. **2)** Then we compare the use of BNs tracked during training. Consistent with the prior study (Li et al., 2020b), locally tracked BN performs better than the globally averaged one. **3)** Rescaling layer outputs barely affect the accuracy, but it could be poisonous for tracked BN statistics. **4)** The initialization rescaling greatly improves the performance regardless of the choice of BN statistics.

In conclusion, either locally tracked or batch averaged BN statistics can yield both efficient and accurate performance. Post-averaged BN statistics may be preferred if post averaging is bearable for efficiency, especially for large-scale datasets. Rescaled initialization is an essential ingredient for Split-Max to perform well. Layer rescale is not recommended if batch average or tracked BN statistics are utilized.

Table 4: Ablation study of network scaling and BN statistics on Digits dataset. Accuracy of different customized widths are presented.

| BN stat | rescale init | rescale layer | ×0.125 | ×0.25 | ×0.5 | ×1 |
|---|---|---|---|---|---|---|
| batch average | ✗ | ✗ | 81.1% | 84.3% | 86.2% | 87.3% |
| | ✗ | ✓ | 81.1% | 84.2% | 86.2% | 87.2% |
| | ✓ | ✗ | 84.5% | 87.5% | 88.9% | 89.8% |
| | ✓ | ✓ | 84.6% | 87.5% | 89.0% | 89.8% |
| post average | ✗ | ✗ | 81.2% | 84.5% | 86.2% | 87.4% |
| | ✗ | ✓ | 81.2% | 84.4% | 86.2% | 87.3% |
| | ✓ | ✗ | 84.5% | 87.5% | 89.0% | 89.9% |
| | ✓ | ✓ | **84.9%** | **87.8%** | **89.3%** | **90.2%** |
| tracked | ✗ | ✗ | 79.6% | 82.8% | 84.8% | 85.7% |
| | ✗ | ✓ | 9.4% | 10.6% | 10.6% | 10.6% |
| | ✓ | ✗ | 83.5% | 86.4% | 87.9% | 88.7% |
| | ✓ | ✓ | 8.8% | 8.8% | 8.8% | 10.6% |
| locally tracked | ✗ | ✗ | 81.1% | 84.3% | 86.3% | 87.3% |
| | ✗ | ✓ | 9.8% | 10.6% | 10.1% | 11.7% |
| | ✓ | ✗ | **84.9%** | 87.7% | 89.1% | 90.0% |
| | ✓ | ✓ | 10.5% | 10.6% | 12.1% | 11.1% |

## A.4 EXPERIMENTS WITH I.I.D FL

In addition to non-*i.i.d* FL settings, we experiment with *i.i.d* FL where each client will own data of 10 classes from the CIFAR10 dataset. Results using 100% and 50% training data are included in Table 5. We observe a great increase in the accuracy compared to the non-*i.i.d* experiments, which is a common phenomenon that non-iid FL will perform worse globally. The similar performance degradation was observed in (Diao et al., 2021), as well. In Table 5, our method performs better in larger widths with fewer training data, whose performance approaches that of unconstrained individual FedAvg. Our method provide a monotonous relation between model size and accuracy (larger models are more accurate) and uses fewer parameters and MACs even compared to wider baseline networks, though performs worse in smaller widths because the slimmest networks are updated less frequently in budget-insufficient clients compared to the SHeteroFL or FedAvg. Worth to mention, our method uses much fewer parameters and operation counts for the same accuracy. For example, Split-Mix requires 7.2M MACs and 1.4M parameters for 81.1% accuracy while SHeteroFL needs twice of the complexity, given the 50% CIFAR10 *i.i.d* configuration.

Table 5: Test results of customizing model width on the class non-*i.i.d* CIFAR10 dataset.

| width | Individual FedAvg | | | SHeteroFL | | | Split-Mix (ours) | | |
|---|---|---|---|---|---|---|---|---|---|
| | Acc | MACs | #Params | Acc | MACs | #Params | Acc | MACs | #Params |
| | CIFAR10 *i.i.d* FL (100%) | | | | | | | | |
| ×0.125 | **82.2%** | 0.9M | 0.2M | 81.9% | 0.9M | 0.2M | 80.9% | 0.9M | 0.2M |
| ×0.25 | 86.1% | 3.5M | 0.7M | **85.2%** | 3.5M | 0.7M | 83.4% | **1.8M** | **0.4M** |
| ×0.5 | 89.8% | 14.0M | 2.8M | **86.5%** | 14.0M | 2.8M | 85.2% | **3.6M** | **0.7M** |
| ×1 | 91.0% | 55.7M | 11.2M | 85.9% | 55.7M | 11.2M | **86.0%** | **7.2M** | **1.4M** |
| | CIFAR10 *i.i.d* FL (50%) | | | | | | | | |
| ×0.125 | **77.3%** | 0.9M | 0.2M | 77.2% | 0.9M | 0.2M | 74.2% | 0.9M | 0.2M |
| ×0.25 | 79.9% | 3.5M | 0.7M | **79.7%** | 3.5M | 0.7M | 77.9% | **1.8M** | **0.4M** |
| ×0.5 | 83.2% | 14.0M | 2.8M | **80.1%** | 14.0M | 2.8M | 79.5% | **3.6M** | **0.7M** |
| ×1 | 84.6% | 55.7M | 11.2M | 75.5% | 55.7M | 11.2M | **81.1%** | **7.2M** | **1.4M** |

## A.5 MORE BUDGET DISTRIBUTIONS

In our experiments, we generally use an exponential budget distribution: $R_k = (1/2)^{\lceil 4k/K \rceil}$. Though the distribution represents the imbalance between the budget-sufficient and budget-insufficient clients, real-world applications may encounter a wider variety of budget distributions. Thus, we extend our problem assumption to budget distributions with *more budget-sufficient clients* where we let more groups to have ×1 or ×0.5 net training capability and with *step-increase budgets* where we increase budgets by a fixed step (e.g., ×0.25). In addition, we consider a *log normal* distribution which concentrate around 0.45 budget with few wider or extremely budget-insufficient clients. We partition the budget distribution into 0.125-width bins and each client will only train the maximal compatible width varying from 0.125 to 1, which greatly increases the number of slimmable subnetworks (8 now compared to previous 4). To reduce the overhead of slimmable training, we use HeteroFL instead of SHeteroFL. Fig. 11 reports the per-width accuracy for Split-Mix and SHeteorFL on the Digits dataset. For SHeteroFL, we only report the evaluated performance on trained widths. In other words, if the maximal width is ×0.5, we will not report results of the ×1 net. For Individually-trained FedAvg (Ind. FedAvg), models individually trained for each width are reported, which ignores the width constraints by users and therefore only serves as reference upper bounds. Regardless of the budget distributions, for instance, more budget-sufficient clients (Fig. 11a) or non-exponential distributions (Fig. 11b), Split-Mix outperforms SHeteroFL with larger widths.

## A.6 EFFECT OF LOWER CONTACT RATES

Because of varying communication conditions in deployment, the times that clients actively and successfully upload their models could be fewer than expected. To evaluate the robustness of customization federated algorithms, we conduct federated experiments with a varying number of active clients per round in Fig. 12. Because of the limited number of communication rounds (within

300 rounds), the test accuracy decreases by fewer contact clients. This is a common phenomenon because lower contact rates requires more communication rounds to reach the same performance as the full-contact competitors do. Though global performance generally decreases, we find that the wider networks are more accurate if trained by SplitMix. One source for the advantage is the modular base models in SplitMix, which can be easily distributed into different rounds for training. Rather, the wider integrated networks in SHeteroFL lower their chance to be aggregated globally and therefore their global performance declines.

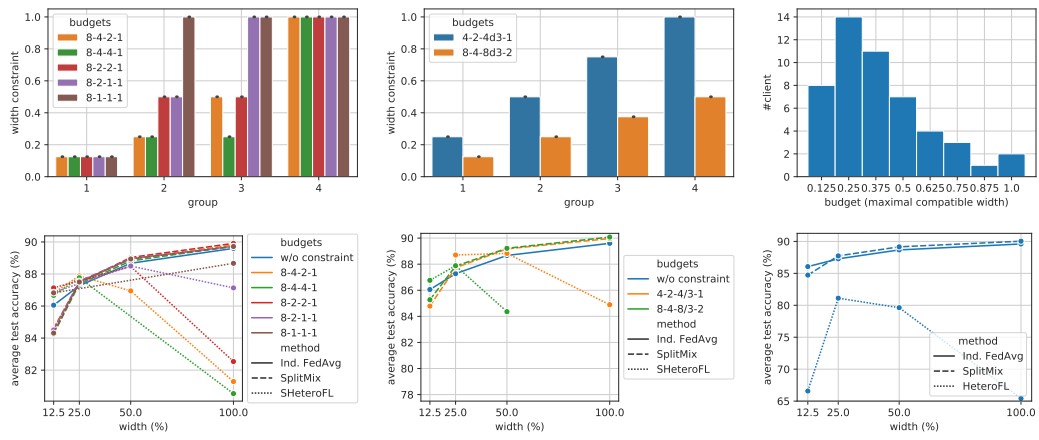

(a) More budget-sufficient clients  (b) Step-increase budget distributions  (c) Log-normal distributions

Figure 11: Vary the budget distribution. The training budgets, i.e., width constraints, are depicted in the upper figures by group. The budget distribution name, for example, 8-4-2-1, means $\times 1/8$, $\times 1/4$, $\times 1/2$ and $\times 1$ width constraints for each group, respectively. The lower figures compare the performance of trained models with customized withs.

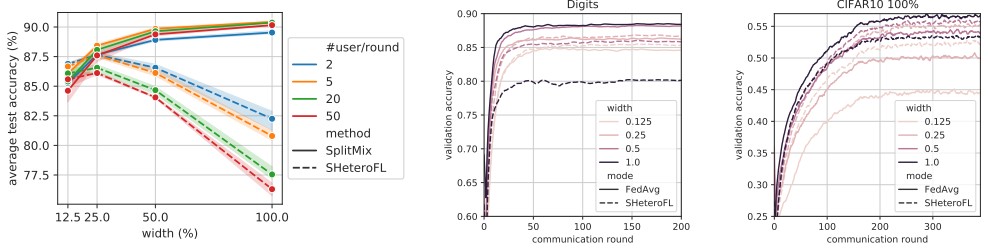

Figure 12: Vary the number of clients uploading models per round.

Figure 13: Convergence of different-width models.

## A.7 NEGATIVE IMPACT OF CONSTRAINED BUDGETS ON WIDER NETWORKS

In Fig. 2, we show how the SHeteroFL fails to train wider networks with budget-constrained clients. To show the generality of such a problem, we extend the experiments to Digits and CIFAR10 datasets in Fig. 13. Though wider networks converges faster at the beginning, they meanwhile overfit limited data in a few clients and therefore their validation accuracy no longer improves.

