# OpenReview forum: "Efficient Split-Mix Federated Learning for On-Demand and In-Situ Customization"
_ICLR.cc/2022/Conference — ICLR 2022 Poster_

### Official Review · Reviewer_emNn · 2021-10-24

**Correctness:** 2
**Technical Novelty And Significance:** 3
**Empirical Novelty And Significance:** 2
**Recommendation:** 3
**Confidence:** 3

**Main Review:**

### Strengths

[Problem / Motivation]

The problem of heterogeneous resources in federated training is of utter importance and very challenging both from the systems and algorithms perspective.

[Related Work]

The authors have a valuable overview of the related work.

[Novelty]

The paper introduces some interesting ideas based on existing work (HeteroFL and adversarial robustness).

### Weaknesses

[Significance]

- The evaluation results are heavily based on the assumption of exponentially distributed budgets.
 Justification is needed for why this distribution is the only one to consider. Otherwise other distributions need to be evaluated to show the generality of the proposal.

- The notion of budget is not clearly defined and potentially problematic. Each client has a budget in terms of physical resources (e.g., latency, memory, CPU time, disk). This budget can translate into the resources necessary for training a network of a specific size (e.g., one x0.5 net). According to line 5 of Algorithm 3, the authors indicate that the same client can train a linear combination of smaller networks that sum to the budget (e.g., x0.5 -> one x0.25, one x0.2, one x0.05). This is not valid as the resource consumption of a network does not scale linearly with its size (at least for some of the important physical resources). This assumption on the relation between network size and budget essentially permits Split-Mix to train all base models and outperform SHeteroFL which can train only a subset.


[Clarity]

Some arguments of the paper are not explained.

- The authors mention: "A naive solution is to simultaneously train multiple models with wanted architectures and properties, which however may lead to prohibitive training cost and latency of switching models." The procedure of switching and why it incurs prohibitive latency costs is not clear.

- In Figure 2: x1 net SHeteroFL seems to converge faster than slimmer nets till around round 50. What happens afterwards ? The authors conclude that the widest model may not be a good candidate model but don’t provide any intuition behind this. After reading further the reason, behind this seems to be the less amount of data.


[Evaluation]

Some results are questionable / counter-intuitive.

- Table 1: Why does the accuracy of SHeteroFL not monotonically increase with FLOPS and #Params (as it is for Split-Mix) ?

- Figure 9b: Why λ=0 gives less SA than λ={0.1, 0.2} ?


[Presentation]

Paper presentation needs improvement.

- The setup for Figure 2 is not clear enough for understanding the demonstrated behaviour. The reader has to read Section 5 to understand the setup and then continue to Section 4. This breaks the flow of the paper.

- Gradient saliency should be defined or described in a few lines.

- Please mention “adversarial robustness” instead of “robustness” at least once in the beginning of the paper.

- There is a large number of typos (mainly after Section 4). A sample:
  - 4.1 (beginning): “To motivation” -> “For motivation”
  - “convergence slower” -> “converge slower”
  - “To validation” -> “For validation”
  - “Last, it worth note that” -> “Lastly, it is worth noting that”



**Summary Of The Paper:**

The paper proposes a new federated learning scheme that is suitable for devices with heterogeneous resources. The proposal, namely Split-Mix, trains multiple models of different sizes and adversarial robustness levels, tailored to the budget of each device. Empirical results demonstrate the efficacy of the method against the main competitor.

**Summary Of The Review:**

I recommend this paper to be rejected mainly given the issues regarding significance and lack of proper explanations for various claims throughout the paper (see Main Review section).

---

> ### Author Response · Authors · 2021-11-21
> **Response to Reviewer emNn on significant concerns**
>
> We appreciate that you point out the importance, challenges, and novelty of our paper. We have tried our best to address your concerns in two posts: for significant ones and detailed questions, respectively.
>
> > The evaluation results are heavily based on the assumption of exponentially distributed budgets. Justification is needed for why this distribution is the only one to consider. Otherwise other distributions need to be evaluated to show the generality of the proposal.
>
> With the exponential budget distribution, we aim to create a representative benchmark that can simulate the challenges of the novel problem in practice:
>
> * *Diverse budgets*: the distribution provides various client budgets with the significant difference between the most efficient and least efficient devices.
> * *Unbalanced budget distribution*: The exponentially distributed budgets are unbalanced where only 25% of users can afford the $\times 1$ net and therefore hardens the task.
>
> Additionally, due to the wider variety of budget distributions in practice other than the exponential distribution, we extend our experiments to budget distributions (1) with *more budget-sufficient clients* where we let more groups have $\times 1$ or $\times 1/2$ net training capabilities and (2) with *step-increase budgets* where we increase budgets by a fixed step (e.g., $\times 0.25$).
> In our revision, Fig. 16 reports the per-width accuracy for SplitMix and SHeteorFL.
> With these budget distributions, for instance, more budget-sufficient clients (Fig. 16a) or non-exponential distributions (Fig. 16b), Split-Mix outperforms SHeteroFL with larger widths and the performance is close to the unrealistic upper bounds without constraint.
>
> > The notion of budget is not clearly defined and potentially problematic.
> > Each client has a budget in terms of physical resources (e.g., latency, memory, CPU time, disk). This budget can translate into the resources necessary for training a network of a specific size (e.g., one x0.5 net). According to line 5 of Algorithm 3, the authors indicate that the same client can train a linear combination of smaller networks that sum to the budget (e.g., x0.5 -> one x0.25, one x0.2, one x0.05). This is not valid as the resource consumption of a network does not scale linearly with its size (at least for some of the important physical resources). This assumption on the relation between network size and budget essentially permits Split-Mix to train all base models and outperform SHeteroFL which can train only a subset.
>
> We defined the notion of budgets in our problem setting. Specifically, in the 2nd paragraph of Sec. 3, each client's computation budget is confined by the maximal trainable network width.
> Such a definition follows HeteroFL (Diao et al., 2021) which is the state-of-the-art FL method to train with different computation capabilities. Therefore, we believe the comparison is fair for both SHeteroFL and Split-Mix.
> We also appreciate your suggestions on other forms of budget definitions, like CPU time or latency. But considering these budgets may be beyond the scope of this paper and could complexify the comparison with HeteroFL (which does not consider such constraint and may perform worse).
>
> To reiterate the definition of budgets, we consider two forms in terms of memory/disk consumption during training.
>
> * *Memory for storing the model*, which scales by the number of parameters.
> We agree with the reviewer that the scaling relation is non-linear but **quadratic**, for which two x0.25 nets should fit into a x0.5-width-capable client.
> Exemplified by a linear layer with weight matrix $\theta \in \mathbb{R}^{d_1\times d_2}$, the scaling relation indicates that two x0.25 networks consumes $2\times (0.25)^2 d_1 d_2=0.125d_1d_2$ parameters, which can definitely fit into a client of x0.5 budget which allows $0.5^2d_1 d_2=0.25 d_1d_2$ parameters.
> * *Memory for storing intermediate layer activations*, which depends on the model widths. For example, forwarding during training requires temporarily storing the representation extracted by former layers of a deep neural network.
> The number of channels (width) explicitly scales up the memory consumption of the intermediate variables **linearly**. Thus, 2 $\times 0.25$ nets (equal to 0.5 width) can fit into a 0.5-width-capable client.
>
> As a result, we argue that our assumption is fair for all baselines.
> Actually, when we fit two x0.25 nets (12.5% of budget parameters) into x0.5 budget, the SHeteroFL trains more parameters (100% of budget parameters) but performs worse than ours.

---

> > ### Comment · Reviewer_emNn · 2021-11-23
> > **Paper recommendation after authors' response**
> >
> > I would like to thank the authors for responding to my review and for revising / improving the paper.
> > The response addressed some of my questions whereas my main concerns remain.
> > In particular a few comments:
> >
> > > The loading/unloading will therefore significantly increase the time complexity of training.
> >
> > Depending on the architecture, the cost for loading a new model into the main memory can be negligible compared to the cost to train the model.
> >
> > > As we demonstrated and discussed in Sec 4.1 and Fig. 2, the wider networks trained by SHeteroFL will be less generalizable due to the limited data samples. In contrast, slimmer networks (x0.125 or x0.25 nets) are trained by more clients/samples (over 3/4 of the whole dataset). Therefore, the accuracy of SHeteroFL will first increase (because of more parameters) and then decrease (because of limited data).
> >
> > I would recommend adding at least one larger dataset (e.g., based on ImageNet) to verify this hypothesis.
> >
> > > A minor adversarial noise resembles mild data augmentation and may help the generalization of the model.
> >
> > Similarly, evidence for this hypothesis would be useful.
> >
> > > Budget distributions
> >
> > The updated setup and experiments are interesting but I would recommend a more thorough explanation (with references and/or empirical evidence) regarding the choice of these distributions.
> >
> > > We agree with the reviewer that the scaling relation is non-linear but quadratic [..] The number of channels (width) explicitly scales up the memory consumption of the intermediate variables linearly.
> >
> > The theoretical estimation of this relation is not adequate for a paper aiming for a practical contribution. If this assumption holds for the majority of the target hardware (which I highly doubt) then the authors should justify it with concrete empirical evidence.
> >
> > ------------------
> >
> > Overall I appreciate the authors effort to address various concerns but believe the paper needs significant improvements and will thus not change my recommendation.

---

> > > ### Author Response · Authors · 2021-11-27
> > > **Response to Reviewer emNn's follow-up questions**
> > >
> > > We appreciate that the reviewer has read our long response and below are our responses to the follow-up questions.
> > >
> > > > The loading time and time complexity of training multiple networks (of different widths).
> > >
> > > We apologize for the confusion caused by our statements. Actually, because of the loading/unloading, the training has to be sequentially done, which significantly increases the training time. For example in Fig. 15, SHeteroFL with sequential training significantly increases the wall time of training and the time is longer than either FedAvg or Split-MIx methods.
> > >
> > > > Larger dataset is recommended to justify non-monotonical accuracy by width.
> > >
> > > We would like to add the results on ImageNet in our revision. However, we argue that our current experimental results on small or large datasets are strong evidence of the limitation of previous methods.
> > >
> > > In three datasets (Fig. 2 for DomainNet, Fig. 18 for Digits and CIFAR10), we observe the non-montonical phenomenon repeatedly: 71.3% (x0.125 net) -> 81.8% (x0.5) -> 81.2% (x1) on CIFAR10, 84.4%(x0.125) -> 86.7%(x0.25)->81.6% (x1) on Digits and 59.4%(x0.125) -> 62.3% (x0.25) -> 56.2% (x1) on DomainNet.
> > > Note that DomainNet has the same or larger image size (255x255) as those used in ImageNet (typically 224x224 after cropping) and there are around 2000 samples per class and 6 non-i.i.d. domains.
> > > Even if we experiment on a TinyImageNet dataset with 10 classes per client (class non-iid) and 100 clients, we still observe the non-monotonical accuracy: 18.2% (x0.25 net) -> 22.4% (x0.5 net) -> 22.1% (x1 net).
> > > The above datasets were widely used in FL with heterogeneous architectures (e.g., HeteroFL by Diao et al., 2021 ICLR), and cross-domain FL (FedBN by Li et al., 2020 ICLR), or other heterogeneous federated learning (FL) (Lin et al., 2020 NeurIPS).
> > > We believe that these authors have common sense as us that the benchmarks can demonstrate the effectiveness of FL algorithms in different aspects.
> > > With these datasets, we can justify that the phenomenon is not a special case by design but generally exists in multiple federated scenarios.
> > >
> > > > Evidence for the minor adv. noise improving accuracy.
> > >
> > > Tsipras et al. (2019) [A] has observed that adversarial training has a positive effect on model generalization especially when training data are limited (for example, MNIST in their Fig. 1). In our Digits experiments, we used 30% of the whole dataset and distribute them to 50 clients. As a result, each client will have limited data, and the adversarial noise in reverse improves the generalization.
> > >
> > > * [A] Tsipras, D., et al. (2019). Robustness May Be at Odds with Accuracy. ICLR
> > >
> > > > A more thorough explanation regarding the choice of budget distributions.
> > >
> > > We choose these distributions to reveal the generality of the experiment results.
> > > In practice, it is hard to predict what kind of clients will participate in the federated learning.
> > > Therefore, we would like to evaluate as many kinds of budget distributions as possible.
> > > The selected distributions have a varying number of clients with sufficient budgets and some clients are very inefficient such that only training a x0.125 net is possible.
> > > In the extended experiments, we also consider when the inefficiency is scarce in clients, for example, 75% clients are sufficient in budgets, or the minimum width is larger than x0.125.
> > > We believe such an evaluation has covered many practical scenarios.
> > >
> > > > The theoretical estimation of budget constraints is not adequate. The authors should justify it with concrete empirical evidence.
> > >
> > > The ‘theoretical estimation’ (or quantitive analysis) provides a general evaluation of the rationality of the assumption, without additional assumptions on the uncertain deployed environments.
> > > Since our work aims for addressing practical problems, we believe a general analysis is important for the method to be effective in different circumstances, unless we missed some special cases.
> > >
> > > To provide a concrete justification, we measure the memory costs of clients at runtime on a CUDA-empowered GPU.
> > > Consider a client who is capable of training a single x1 net by FedAvg.
> > > The running memory allocated by the local training program will be 130 Mb.
> > > In comparison, a SHeteroFL client will require 129.5 Mb and a SplitMix client requires 129.5 Mb, too.
> > > The allocated memory is measured by CUDA API at runtime, which is the peak memory cost during training including auxiliary temporary variables and runtime costs.
> > > The cost of SplitMix is reported by the inefficient implementation: embedding base models into an integrated model for parallel training (c.f. Fig. 14c).
> > > Because of the implementation, SplitMix costs more memory than our theoretical analysis.
> > > Though with minor differences in the memory cost, our method does not break the budget constraint at running.
> > > Even though with slimmer budgets, the memory costs comparisons are similar between SplitMix and individual FedAvg: 32.0 vs 33.9 (x0.5), 8.1 vs 8.1 (x0.25) and 2.1 vs 2.1 (x0.125).

---

> > > > ### Comment · Reviewer_emNn · 2021-11-29
> > > > **Thank you for the follow-up**
> > > >
> > > > I thank the authors for the follow-up comments that contain some additional information.

---

> ### Author Response · Authors · 2021-11-21
> **Response to Reviewer emNn on clarity/evaluation/presentation**
>
> > The authors mention: "A naive solution is to simultaneously train multiple models with wanted architectures and properties, which however may lead to prohibitive training cost and latency of switching models." The procedure of switching and why it incurs prohibitive latency costs is not clear.
>
> Given a budget-constrained device, we assume that the maximal loadable parameters and memory footprint (e.g., the size of layer activations) are limited.
> Therefore, naively training multiple architectures with non-overlapped parameters needs to repeatedly unload a network before loading and training another network of different widths/architectures.
> The loading/unloading will therefore significantly increase the time complexity of training.
>
> > In Figure 2: x1 net SHeteroFL seems to converge faster than slimmer nets till around round 50. What happens afterwards? The authors conclude that the widest model may not be a good candidate model but don’t provide any intuition behind this. After reading further the reason, behind this seems to be the less amount of data.
>
> We discussed the case in the 3rd paragraph of Sec 4.1, *following which we provide the intuition* (in our initial submission).
> The stated intuition is consistent with the reviewer's opinion:
> "Intuitively, as a consequence of the budget constraint, fewer training clients are able to participate in training wider networks, where the reduced sample size or model size greatly impacted the generalization."
>
> > Table 1: Why does the accuracy of SHeteroFL not monotonically increase with FLOPS and #Params (as it is for Split-Mix)?
>
> As we demonstrated and discussed in Sec 4.1 and Fig. 2, the wider networks trained by SHeteroFL will be less generalizable due to the limited data samples.
> In contrast, slimmer networks (x0.125 or x0.25 nets) are trained by more clients/samples (over 3/4 of the whole dataset). Therefore, the accuracy of SHeteroFL will first increase (because of more parameters) and then decrease (because of limited data).
>
> > Figure 9b: Why $\lambda$=0 gives less SA than $\lambda$={0.1, 0.2} ?
>
> A minor adversarial noise resembles mild data augmentation and may help the generalization of the model.
>
> > The setup for Figure 2 is not clear enough for understanding the demonstrated behaviour. The reader has to read Section 5 to understand the setup and then continue to Section 4. This breaks the flow of the paper.
>
> With Fig 2, we aim to demonstrate the awful generalization of wider networks by SHeteroFL due to the limited data.
> Thus, we stated the key setup for Fig 2, the imbalance budgets, which confine the training of wider networks in a few clients/data (see the 3rd paragraph of Sec. 4.1).
> To keep the focus of the discussion on motivation in Sec. 4.1, we delay other details of the experiments, like model architectures or data distributions, to Sec. 5.

---

### Official Review · Reviewer_82wJ · 2021-10-28

**Correctness:** 3
**Technical Novelty And Significance:** 3
**Empirical Novelty And Significance:** 3
**Recommendation:** 8
**Confidence:** 3

**Main Review:**

## Strengths
- Addressing data and resource heterogeneity is definitely one of the important topics of FL.
- The main technical idea, which splits the global network into a collection of smaller base models of various sizes (though I do not perfectly understand how it was done actually) and asks clients to train a random subset of them under computational resource constraints, seems like a straightforward and reasonable solution to the problem.
- Also, jointly training both accurate and robust models while keeping all but BN layers shared sounds like an interesting idea, though it could be useful in contexts other than federated learning.
- Experiments are thorough, demonstrating the effectiveness of the proposed approach on multiple datasets with multiple metrics.

## Weaknesses
It's not perfectly clear how the split and aggregation are done in the proposed approach. On the one hand, Figure 1 illustrates that multiple base models are aggregated into literally a single global model, which is split into random base models that may or may not have identical architectures over rounds. On the other hand, Algorithm 3 describes that W^t is given by a set of base models, which are not aggregated into a single model but kept as a set of models throughout the training procedure. Which is the one that the proposed approach is actually doing?

I think that clarifying the above point is important as this would be one of the major differences between the proposed approach and HeteroFL. HeteroFL aggregates multiple models with different sizes into a single one, and SHeteroFL asks clients to train all affordable base models. If the SplitMix does not actually aggregate multiple base models, does this the main reason to bring performance improvements in the experimental results?

Another concern is the problem setting. The proposed work argues that "Without loss of generality, we assume exponentially distributed budgets in uniformly divided client groups:..." I wonder how much this assumption is reasonable and how dividing clients into four groups in this way is optimal for various situations. My simple question is: what happens if we change the number of groups to better deal with the computational resource heterogeneity of clients, and what happens when clients are not distributed in an assumed way regarding the computational resource (e.g., many clients have sufficient computational budgets)? Further clarification on this point would make the experimental results stronger.

**Summary Of The Paper:**

This paper presents a new federated learning approach named Split-Mix FL that allows clients to train customized models efficiently while considering heterogeneity in data and computation resources. The key idea is threefold: 1) the global model is first split into several sub-networks called base models that each have different sizes, thus requiring a different amount of computational resources; 2) Each selected client trains a random subset of base models, under its computational resource constraints; 3) updated base models are aggregated at the server-side and distributed again. Furthermore, the proposed approach can train both accurate and robust models in a joint fashion, where all but batch-norm layers are shared for efficiency. Experimental results on multiple datasets (CIFAR10, Digits, DomainNet) demonstrate the effectiveness of the proposed approach compared to FedAvg and HeteroFL.

**Summary Of The Review:**

Overall, I think that this is a nice contribution with compelling empirical results. Although the manuscript is largely well written, it was not clear to me how model split and aggregation are done in the proposed approach, and HeteroFL, SHeteroFL, and Split-Mix are different and how these differences bring performance improvements. Also, the assumption about the computational resources of clients would be better clarified.

---

> ### Author Response · Authors · 2021-11-21
> **Response to Reviewer 82wJ**
>
> We appreciate the reviewer acknowledging the novelty and importance of our work. For the raised concerns, we have tried our best to address them in revision and posts here.
>
> > Weaknesses
> >
> > It's not perfectly clear how the split and aggregation are done in the proposed approach. Which (Fig. 1 or Alg.3) is the one that the proposed approach is actually doing?
>
> Both Alg. 3 and Fig. 1 illustrate that the global model is split into models with homogeneous architecture but different parameters. Due to the raised concern, we updated the confusing illustrations in our revision. Details are provided below.
>
> By Fig. 1b, the global model is split into a set of base models, which have the **same architecture but different parameters**.
> For example, in the 2nd layer of split networks, all models have one node (*same architecture*) but the position of the node is different by different bases (*different parameters*).
> Also described in Alg. 3, these base models are sent to clients randomly depending on the client’s capability.
>
> Besides, in each client, we do NOT aggregate models into a single model.
> The 'aggregated' client models in Fig 1b are supposed to represent the relation of different base models: They work on the same input and output dimensions.
> To clarify, the first layer and the last layer represent *the input sample (image $x$) and output prediction (label $y$)*, respectively.
> Regarding the confusion, we revise Fig 1 and add explicit legends to highlight the two kinds of nodes.
> In addition, we illustrate the spliting and mixing in Fig. 14 by the weight matrix of a layer, which may ease the understanding of our principle.
>
> > If the SplitMix does not actually aggregate multiple base models, is this the main reason to bring performance improvements in the experimental results?
>
> The splitting is one of the key components of our idea and necessarily distinguished our method from (S)HeteroFL.
> In principle, the splitting enables all parameters in a mixed wide model to be trained over all clients.
> In contrast, the wide models in SHeteroFL have a large portion of parameters that are under-fitted.
> For example, 75% parameters of x1 net is only trained on 25% data/clients constrained by the exponentially distributed budgets. Detailed comparison of the portion of trained parameters by the client is provided in Fig. 8, which empirically explained the success of our Split-Mix principle.
>
> > Another concern is the problem setting. The proposed work argues that "Without loss of generality, we assume exponentially distributed budgets in uniformly divided client groups:..." I wonder how much this assumption is reasonable and how dividing clients into four groups in this way is optimal for various situations.
>
> We agree with the reviewer but we did not mean to claim that the dividing is optimal for various situations.
> Instead, we want to create a representative benchmark that can simulate the challenges of the novel problem in practice especially in the presence of the non-iid features.
> Specifically, uniformly dividing users into 4 groups can distribute the budgets into different domains, since we also divide users into domains in order in our experiments (see Fig. 8).
> For example, the x0.125-width clients are mainly in the first two domains of Digits datasets while x1-width clients are in the last two domains.
> Therefore, merely training a x1 net model in one or two domains may not generalize to the rest domains, as we demonstrated in Fig. 8.
> To clarify our points, we update our statements at the beginning of page 4 accordingly.
>
>
> > My simple question is: what happens if we change the number of groups to better deal with the computational resource heterogeneity of clients, and what happens when clients are not distributed in an assumed way regarding the computational resource (e.g., many clients have sufficient computational budgets)?
>
> Thanks to your suggestions, we realize there could be a wider variety of budget distributions in practice other than the exponential distribution.
> Therefore, we extend our experimental assumptions to budget distributions with *more budget-sufficient clients* where we let more groups have $\times 1$ or $\times 0.5$-width capabilities.
> In our revision, Fig. 16 reports the per-width accuracy for SplitMix and SHeteorFL.
> With the diverse budget distributions, SplitMix is relatively stable compared to SHeteroFL and performs better in larger widths.

---

> > ### Comment · Reviewer_82wJ · 2021-11-23
> > **Thanks**
> >
> > Thank you very much for the response. I think my initial concerns have been mostly addressed. In particular, the additional experimental results on various budget distributions (Fig 16 in the appendix) were nice. I'll keep my original rating.

---

> > > ### Author Response · Authors · 2021-11-23
> > > **Thanks to Reviewer 82wJ**
> > >
> > > Dear Reviewer 82wJ,
> > >
> > > Many thanks for reading our responses and letting us know your thoughts.
> > >
> > > Best wishes,
> > > Authors

---

### Official Review · Reviewer_7P9j · 2021-11-02

**Correctness:** 3
**Technical Novelty And Significance:** 2
**Empirical Novelty And Significance:** 2
**Recommendation:** 3
**Confidence:** 4

**Main Review:**

### Strong and Weak points

- (strong) while most works aim at adapting the compute/memory of a network to match the capabilities of the target client, Split-Mix also considers model robustness. This is an important consideration.

- (strong) the Authors do a good job with Figure 2, showing that the widest model might not always be a good fit (i.e. it might overfit if the data is skewed or contains very few training examples)

- (doubt) the Authors stress about the `in-situ customization` of their proposed framework. As they state, customization is possible "once training is done". My doubt is, in what context is customization useful after training? is it do do inference using the learned based models? is customization available if another FL task (of the same type -- e.g. image classification) is going to start after Split-Mix is done?

- (comment) the Authors seem to state that locally training multiple models in parallel is feasible. Is this a realistic assumption? Can a smartphone or a NVIDIA Jetson device really train (which comes with a relatively high memory peak requirement and energy consumption) several models in parallel ?

- (doubt) if all base models need to be trained on all clients, wouldn't this translate into a slow FL training process? It would be interesting to see the line plots in Figure 6 showing wall clock time. In 5.1.convergence the Authors state "all base models are independently trained".

- (weak) the `split` stage is not properly explained: how are the base models derived? Even if fixing the ratio to, for example, 0.125x there are many ways of making a model 8x smaller by discarding channels. This is one of the main ideas in the paper but is not covered. Some approaches are better than others, for example, accuracy would be better retained if more channels are discarded towards the last layers (which are also the layers with more parameters) instead of doing so uniformly.

- (weak) the `mix` stage is not properly explained: From the sentence in the introduction "we mix selected base models to construct desired widths and required robustness", it gives the impression that base models might be merged before sending them to a client but the Authors do not describe how this happens. For example, if a device can afford training a 0.25x model, how are 0.125x base models combined into a 0.25x model? I imagine there are different ways to do this depending on whether the base models are "disjoint" (in the full model representation) or overlapped. This is one of the main ideas in the paper but is not covered.

- (weak) "we measure latency in terms of FLOPS". This is wrong. Latency can be estimated using FLOPS (although is known not to be an accurate representation) but Authors never report any latency estimates (there are no training "wall time" or inference "time") in the paper.


- (weak) The number of parameters for ResNet18 x1 is ~11M (not the reported 5.6M). The number of FLOPS for a ResNet18 using as input a 1x3x32x32 tensor (just a single CIFAR-10 image) is 1.1 Billion FLOPs (I believe Authors are reporting MACs, which, in general are = 0.5*FLOPs). The Authors should revise all FLOPs and #params reported in the paper.


- (weak) For non-IID experiments, the Authors split Digits and DomainNet datasets into 10 and 5 clients, respectively. Given such a small number of partitions, does it mean all clients are sampled at all rounds? I do not think this can be considered a realistic FL setting. This conflicts with the argumentation in the paper, aiming at considering more realistic FL settings accounting for the `different dynamics` and heterogeneity.

**Summary Of The Paper:**

Split-Mix is a Federated Learning strategy aimed at easing the problems that arise when a heterogeneous pool of devices/clients (i.e. some with more compute/memory capabilities than others, different data distributions) collaboratively train a global model. Split-Mix trains a model that can later on be customized in terms of model size and robustness. As the name suggests, Split-Mix has two stages: During `split`, a large model is split into smaller base models. Base models are constructed by discarding channels but maintaining the the number and type of layers in the network. During `mix`, the server samples a fraction of the base models (depending on a given client's compute capabilities) and fuses them into a single one that is send to a client to train. All base models are trained on all clients (should these meet the compute requirements of a sub-model) in a federated manner. This means that the more capable devices, train all base models. This is envisioned to happen in a parallel fashion in a given device. Once FL training is completed, a customized model can be deployed to a device/client in hardware-aware and robustness-ware fashion. The Authors refer to this as _in-situ customization_


**Summary Of The Review:**

### Recommendation

The main weaknesses of this work is that is that the contributions are not properly explained, Split-Mix requires training the base models in all clients (which is unrealistic for large FL settings); for ResNet18 the FLOPs and number of parameters reported are not computed correctly for the x1 baseline (the others might be inaccurate too). The Authors, without argumentation why, directly state that FedAvg baselines can only afford training 0.125x models, why is this? Which device cannot afford training a 0.25x ResNet18 ? Even a RaspberryPi 4 (i.e. CPU-only) can train a x1 ResNet18 using batchsize=32, which takes ~15seconds per batch. As the Authors state themselves, the advantages of Split-Mix for CIFAR10 and DomainNet (the two most challenging task tested in this work) at somewhat limited. What the results evidence is that only models that are somewhat over-parameterized for a given task (as seems to be the case for Digits -- only a 4% acc drop when going from x1 to x0.125) work well with the proposed Split-Mix strategy. In addition, the Authors should evaluate other types of client heterogeneity distributions.

Summary: The Authors have motivated well the paper but need to improve the presentation of the technical contribution. A good way of achieve things would be to present Algorithm 3 sooner and then explain its main stages. To me it does not sound natural to give such brief explanations about `split` and `mix` only at the end of Page 6, just before the experiments section.


### Supporting my Recommendation

Please see points above.

### Minor points


- Is it fair to add as cite to "Federated Learning" a paper from 2019 (first sentence of the Introduction) ?
- When describing Fig3 (in Page 4) the sentence starts with "To validation, we investigate [...]". Maybe it should be "To validate this"?
- 6th line in Sec5: "The dataset first is [...]", should be "The first dataset is [...]"
- When Authors mention "layer output cache", I have assumed this is equal to layer activations or "activations buffer". Please correct me if this means something else.
- It is not clear which architecture was used for Digits. Adding a 1-line description of what the task is for Digits and DomainNet would be useful for readers not used to work with these datasets.

---

> ### Author Response · Authors · 2021-11-21
> **Response to Reviewer 7P9j on major concerns**
>
> We thank the reviewer for detailed comments and suggestions. We tried our best to address your concerns as follows.
>
> > the contributions are not properly explained,
>
> As we summarized at the end of Sec. 1, our contribution is three-fold.
> 1) We are the first to study training a customizable model with heterogeneous local budgets.
> 2) We propose a technically novel Split-Mix framework to aggregate heterogeneous clients into a width- and robustness-adjustable model.
> 3) We empirically validated our method through a comprehensive set of experiments, demonstrating its effectiveness under heterogeneous budget constraints and data distributions.
>
>
> > The Authors, without argumentation why, directly state that FedAvg baselines can only afford training 0.125x models, why is this? Which device cannot afford training a 0.25x ResNet18 ? Even a RaspberryPi 4 (i.e. CPU-only) can train a x1 ResNet18 using batchsize=32, which takes ~15seconds per batch.
>
> First, we follow the constrained setting by HeteroFL (Diao et al., 2020), where different clients have different capabilities of training different widths of models.
> Following (Diao et al., 2020), we use the same network, PreResNet18, for a fair comparison. In this setting, FedAvg baselines can only train 0.125x models, because some clients can only afford 0.125x net and a unified network architecture is required for all clients by FedAvg.
>
> In addition, a report (Ignatov et al., 2018) has demonstrated the diversity of device capabilities.
> The inference latency of mobile phones could vary a lot: from 3ms to more than 1 min (Asus Memo Pad 7).
> Since the complexity of training is times of the inference, we can easily conclude the much higher latency in training.
> Regardless of the latency, phones like LG Optimus L7 only have a small RAM as 500MB.
> Given such a big latency and limited memory, it is irrational for phone users to sacrifice time or space of the normal use for training a model, especially when mobile phones are the common devices in FL (McMahan et al., 2017).
>
> (Ignatov et al., 2018) Ignatov, A., Timofte, R., Chou, W., Wang, K., Wu, M., Hartley, T., & Van Gool, L. (2018). AI Benchmark: Running Deep Neural Networks on Android Smartphones. CVPR. https://ai-benchmark.com/ranking
>
>
> > As the Authors state themselves, the advantages of Split-Mix for CIFAR10 and DomainNet (the two most challenging task tested in this work) at somewhat limited. What the results evidence is that only models that are somewhat over-parameterized for a given task (as seems to be the case for Digits -- only a 4% acc drop when going from x1 to x0.125) work well with the proposed Split-Mix strategy.
>
> We do not think the reviewer’s comparison is fair regarding the constrained setting in our paper.
> In the constrained setting, the limited local computation capability of 25% clients has eliminated the possibility of directly training wider models individually.
> Therefore, the performance of individually trained models by FedAvg is an unrealistic upper bound.
> The gap between the individual performance and SHeteroFL/SplitMix indeed evidences the difficulties of the proposed novel problem setting.
> Even in such a difficult constrained setting, our method yields customizable wider models whose inference accuracy is promoted by loading more trained parameters, as shown in Table 1.
>
> > In addition, the Authors should evaluate other types of client heterogeneity distributions.
>
> In Appendix B.2 (of revision), we experiment with class non-i.i.d settings, where each client will randomly select 3 out of 10 classes from the CIFAR10 dataset.
> Results are included in Table 4.
> We observe a great drop in the accuracy compared to the i.i.d experiments.
> It is a common phenomenon that non-iid FL will perform worse globally. Similar performance degradation is observed in (Diao et al., 2021).
> Besides the degradation, our method outperforms baselines consistently and its performance is comparable to the unconstrained individual FedAvg with wider customizations.

---

> ### Author Response · Authors · 2021-11-21
> **Response to Reviewer 7P9j on detailed comments of paper weakness**
>
> > (weak) the split stage is not properly explained: how are the base models derived? Even if fixing the ratio to, for example, 0.125x there are many ways of making a model 8x smaller by discarding channels. ...accuracy would be better retained if more channels are discarded towards the last layers...
>
> We derive the base models at the initialization stage.
> A demonstration of weight matrices of the base models is given in Fig. 14c, where we uniformly split a single weight matrix along the input/output dimensions.
> As all parameters are randomly initialized (thus they are equivalent) and all base models are independent (not mutually connected), the choice of parameters for base models should not affect the downstream training.
>
> We appreciate the suggestion for discarding more channels in the last layer.
> However, one of our goals is to keep the maximal width of the network, we have to cut off some channels in the intermediate layers (which is typically wider than the last one).
> Adaptive width is a good idea, which should be considered to be applied for improving our current method in the future.
>
> > (weak) the mix stage is not properly explained: From the sentence in the introduction "we mix selected base models to construct desired widths and required robustness", it gives the impression that base models might be merged before sending them to a client but the Authors do not describe how this happens. For example, if a device can afford training a 0.25x model, how are 0.125x base models combined into a 0.25x model? I imagine there are different ways to do this depending on whether the base models are "disjoint" (in the full model representation) or overlapped. This is one of the main ideas in the paper but is not covered.
>
> All base models are disjoint as illustrated in Fig 1 or Fig. 14c.
> In Line 11, we have stated how the $\times R$-width model can be constructed by *combining the outputs* of selected base models.
> So the base models are disjoint and two x0.125 nets should cost less memory than one x0.25 as demonstrated in Fig. 14c.
>
>
> > (weak) "we measure latency in terms of FLOPS". This is wrong.
>
> We follow the common practice of previous customizable networks, e.g., (Yu et al., 2019; Wang et al., 2020; Huang et al., 2018; Wang et al., 2018) to evaluate the latency in terms of FLOPs.
> The evaluation is also used in FL (SHeteroFL by Diao et al. 2021).
> Though not an accurate measurement of latency, FLOPs are independent of the implementation of low-level APIs, or hardware or randomness of executions.
> Given the same network architecture (except the width), FLOPs provide a relative measurement comparing two different methods.
> In comparison, the wall time is sensitive to the execution environment, which may hurt the reproducibility of the results.
>
> Considering the complexity of training, e.g., the parallelism, we report the convergence of three methods versus the wall training time in Fig. 15. We show that our method is much more efficient than SHeteroFL and only slightly slower than the slimmer and under-fitted FedAvg baseline.
>
>
> > (weak) The Authors should revise all FLOPs and #params reported in the paper.
>
> Thank you for the nice catch. Our previously reported parameter numbers are indeed halved, and we updated all results in the revised version.
> Since the miscalculation affects all results almost equally, the comparison conclusions have not been affected.
>
> As we stated in the 2nd sentence of the last paragraph in Page 7, our definition of  FLOPS is given by the number of multiplication-or-add operations, which could be misleading. We followed the suggestion of the reviewer and revised all the mentions to MACs.
>
> > (weak) For non-IID experiments, the Authors split Digits and DomainNet datasets into 10 and 5 clients, respectively. Given such a small number of partitions, does it mean all clients are sampled at all rounds? I do not think this can be considered a realistic FL setting.
>
> The split is domain-wise as we state in the initial version: "**Each domain** of Digits (or DomainNet) is split into 10 (or 5) clients" (the 2nd last sentence of the 1st paragraph in Sec. 5).
> In another word, the Digits dataset will be split into 10 x 5=50 clients and DomainNet is 5x6=30 clients.
> The experiment used the benchmark adopted in the previous paper (FedBN by Li et al., 2020b; Peng et al., 2019b), where all clients are sampled at all rounds.

---

> ### Author Response · Authors · 2021-11-21
> **Response to Reviewer 7P9j on minor comments**
>
> > (doubt) in what context is customization useful after training? ... is customization available if another FL task (of the same type -- e.g. image classification) is going to start after Split-Mix is done?
>
> The customization can be used whenever the inference resource is dynamically changing (calling for width-customization) or the security sensitivity is changing (calling for robustness-customization).
> We have discussed practical applications of such customization in the *3rd paragraph of Sec. 1* and a number of prior work or applications of in-situ model customization including (Wang et al., 2020; Yu et al. 2018; Wang et al. 2018; Huang et al. 2018).
> For example, memory consumptions and computational capabilities may vary drastically between training and inference stages, and the same dynamics in hardware can also happen during inference (Yu et al. 2018; Xu et al. 2021), depending on how the resource allocation of the running programs is prioritized on a participant’s device (e.g., by operating systems).
> Another example of dynamics is the requirements of robustness properties of the model.
> In autonomous driving, raining heavily all of a sudden can immediately and drastically lower the quality of the camera and other sensors, and therefore one would like to quickly switch to a more robust prediction model, as compared to the one used on sunny days.
> The dynamics along with the low latency requirements for the inference, altogether demand a model that can be customized \emph{in situ}, meaning that we can instantly tailor our predictive model to a proper size that fits our available budget, and promptly adjust its robustness.
>
> Yes. The inference is executed with the proper width of the ensemble of learned base models, as we stated in *the last line of Alg. 3*.
>
> Split-Mix can be applied to the downstream FL tasks that an equal-width integrated network can be applied to, because Split-Mix and the integrated networks are similar in structure.
> Though this problem is beyond the scope of the paper, we think this is an important research problem that we will investigate in the future.
>
> > (comment)  Is parallelism of multiple bases a realistic assumption? Can a smartphone afford?
>
> We agree with the reviewer that the parallelism of two x0.25 models requires more memory.
> However, we emphasize that the increase of memory is measured on comparison to a *single* x0.25 base model, which is somehow *unfair*.
> Remarkably, training two x0.25 nets (in Split-Mix) is more memory-efficient than training a single x0.5 net (in HeteroFL), which is a fair competitor.
> Compared to equal-width baselines (as we do in Table 1), Split-Mix does not require more memory but less memory.
> As conceptually illustrated in Fig. 1, given the same constraint, e.g., x0.5 width budget, both the set of two x0.25 base models (of Split-Mix) and the single x0.5 net (of HeteroFL) should fit into the client.
> For example, if an NVIDIA Jetson device can afford an x0.5-width network in training, then it will afford training 2 x0.125-width base networks in parallel.
>
> > (doubt) if all base models need to be trained on all clients, wouldn't this translate into a slow FL training process? Need the line plots in Figure 6 showing wall clock time.
>
> We think the statement 'all base models need to be trained on all clients' may cause confusion without a specific context.
> Indeed, one key component of our idea is to train all base models on all clients, **but NOT in one communication round**.
> As illustrated in Line 5 of Alg 3 or Fig. 14 (c), we only sample a subset of base models to be trained locally *in each communication round*.
> The number of base models to be trained depends on the capable network width $R_k$ of client $k$.
> For example, if the client $k$ is capable of training $\times 0.5$ Net, then we will send randomly selected 4 $\times 0.125$ Nets to the client for local training.
> Therefore, training multiple base models can be done in one pass instead of sequential passes, and is at least as efficient as training an equal-width single network (with a very inefficient parallel implementation).
>
> To justify the above argument, we conduct experiments to show the wall-clock time in Appendix B.1 and Fig. 15.
> In Fig. 15, Split-Mix is much more efficient than the SheteroFL due to parallelism and fewer non-overlapped parameters.
> We note that FedAvg only trains the $\times 0.125$ net which includes much fewer parameters.
> Therefore, Split-Mix is slightly slower than FedAvg, but the slight degradation of efficiency brings in substantial gains in accuracy.
>
> By 'independently trained', we mean the data, initialization, and computation are not overlapped.
> In comparison, SHeteroFL shares parameters among slim and wide networks, which prohibits the parallel training of multiple-width networks. An illustration is provided in Fig. 14 to visualize the non-overlapping by layer weights.

---

> ### Author Response · Authors · 2021-11-27
> **A kind reminder to Reviewer 7P9j**
>
> Dear Reviewer 7P9j,
>
> Thank you for your time to review our paper and leave valuable comments and suggestions. We are wondering whether you have got a chance to read our response to your questions? We will be glad to provide more explanations and answer more questions if you have any.
>
> Authors

---

### Official Review · Reviewer_Yvwo · 2021-11-03

**Correctness:** 3
**Technical Novelty And Significance:** 3
**Empirical Novelty And Significance:** 3
**Recommendation:** 6
**Confidence:** 4

**Details Of Ethics Concerns:**

This paper has no ethic concerns.

**Main Review:**

## Strengths:
1. The author clarifies the heterogeneity under FL scenario into two aspects, resource and data. Then, the importance of model customisation is intuitive and motivates this work.
2. In terms of the idea "Split-Mix", the authors present empirical evidence of slim models and wide models, including convergence and gradient saliency.
3. I like the deep dive in the evaluation section, both the client-wise statistics in Figure 7 and the percentage difference of parameters locally trained between SHeteroFL and Split-Mix. It vividly shows why these approaches perform as presented and how models with different width perform on different categories.

## Weaknesses:
1. The difference between customisable models in a non-FL setting is somehow vague. It is true that the training data are non-sharable. But the primary question of why the mentioned approaches cannot be adapted to the FL scenario could be further clarified. Are there any technical challenges? Or are these approaches tightly coupled with shareable training data?
2. The evaluation could include training time benchmarks of different approaches. As said that Split-Mix allows all base models to be trained on all clients, what is the cost of time in terms of such training method?
3. The idea of BN layer-wise mixing is kind of confusing. The reason of sharing all parameters except the batch-normalisation requires further elaboration.

## Detail Comments:
1. The notations in the three algorithms lack necessary explanation.
2. Regarding the second contribution stated in Sec. 1, it is better to present direct evidence of efficiency in storage, model loading and inference. For example, the time to load the model, the inference time and latency. The existing evaluations like #Params and FLOPs are indirect metrics
2. The two considerations listed in Sec. 3 are somehow similar. It is better to clarify the difference between heterogeneous computational budgets and imbalance computation budgets.
3. The authors discuss their solutions on customising robustness as it is quite essential in federated learning. I would like to know how this idea can be adapted to other aspects like as mentioned in Sec. 3 neural -architecture search and fairness.
4. I am a little confused by the illustration of Fig. 5. So there are two BN in the training process, each of which handles noised/clean data. And in the inference process, the results are aggregated based on the coefficient. How are the noised and clean data differentiated during training? Are the label input together with the training data?
5. Regarding the evaluation setup, it is recommended to have more details on the clients' budget, e.g. the distribution of client budgets. Are there more clients with a large budget? I suppose the distribution will impact the ultimate accuracy performance.
6. Some minor questions. In Fig. 9(a), why is there only one data point of FedAvg+OAT? Is it due to the divergence in training? In Fig. 7, why are there outliers? It is better to have some general description the baseline methods (i.e. OAT in-situ method) in the evaluation section. I am curious about the condition variable. So what is the ultimate setting in the evaluation section?
7. Some minor typos/grammar mistakes. In the **Layer-wise mixing" of Sec. 4.2, "a straightforward solution is averaging their outputs". In the ** Training and valuation** of Sec. 5.2, the abbreviation of"robust accuracy" should be "RA" instead of "SA".

**Summary Of The Paper:**

This paper proposes a customisation strategy named "Split-Max" for federated learning. The authors identify the heterogeneity of devices and data in FL scenarios. They present the importance of considering devices' budgets and dynamics when dispatching training models. Split-max can adjust the model size according to the devices' budget while maintaining good accuracy and robustness.

Split-max works in steps. First, multiple base models from different initialisations are trained to improve diversity. These base models are randomly given to clients to extract generalisable features. Then, base models are aggregated to the server. Secondly, to provide devices with models with different robustness, it trains two similar models together to capture both the standard-training accuracy and adversarial-training accuracy. Then layer-wise mixing is conducted to achieve both standard accuracy and adversarial accuracy.

Experiments show that Split-Mix achieves better accuracy than naive approaches. Moreover, with customisation, the models are smaller and more robust under budget constraints.

**Summary Of The Review:**


This paper proposes to tackle the heterogeneous budget in FL scenario. A weak accept is given based on two reasons.
(1) The Split-Mix not only considers how to customise models on different clients but also includes how to maintain high robustness of client models where they are trained over non iid data. I think the paper is well motivated and the solution proposed is intuitive and practical. Though the technique used to improve robustness (ie.e ST and AT, parameter sharing and layer-wise mixing) is not new, it well suits the problems and the authors have successfully integrated them with Split-Mix. Moreover, the evaluation shows promising results that Split-Mix could train client models with much fewer parameters but have similar accuracy performance as FedAvg under the budget constraints.

(2) As mentioned in the Main Review, more clarification could be provided to improve the claims in the paper. Why can't conventional model customisation apply to FL scenario? How to handle customisation on other aspect? And the evaluation could include more details regarding training setups and efficiency.

---

> ### Author Response · Authors · 2021-11-21
> **Response to Reviewer Yvwo on major concerns**
>
> We appreciate the reviewer acknowledging the value of our work. We have addressed most of your major concerns here and detailed comments in the following post.
>
> > Weaknesses:
> >
> > 1. The difference between customisable models in a non-FL setting is somehow vague. It is true that the training data are non-sharable. But the primary question of why the mentioned approaches cannot be adapted to the FL scenario could be further clarified. Are there any technical challenges? Or are these approaches tightly coupled with shareable training data?
>
> We mentioned two alternative kinds of customizable networks and discussed their **limitation in memory footprint** compared to the slimmable network such that they canot be applied in our constrained setting (see the 2nd paragraph of Sec. 2).
> They typically consume more memory to store the activations in each layer.
> For instance, sub-depth networks (Huang et al., 2018; Wu et al., 2019) has customizable depths but the widths are fixed.
> In comparison, slimmable neural network (SNN) has a flexible channel width for more adaptive memory footprint.
> Moreover, until the paper was written, SNN is the only heterogenous-architecture training method that has been shown to be applicable in federated learning, namely HeteroFL.
>
>
> > 2. The evaluation could include training time benchmarks of different approaches. As said that Split-Mix allows all base models to be trained on all clients, what is the cost of time in terms of such training method?
>
> To show the training efficiency of our method, we plot the convergence curves in Fig. 15.
> Thanks to the parallel training of multiple base models, our method (customizable for x0.125, x0.25, x0.5, and x1 nets) is much more efficient than SHeteroFL and only slightly slower than the FedAvg with x0.125 nets.
> In contrast, SHeteroFL suffers from the sequential training of models of different widths, resulting in large time complexity.
>
> > 3. The idea of BN layer-wise mixing is kind of confusing. The reason of sharing all parameters except the batch-normalisation requires further elaboration.
>
> The idea is motivated by previous work in adversarial training (Xie & Yuille, 2019; Xie et al., 2020).
> Sharing all parameters except the batch-normalization (BN) layers can maximize robustness and accuracy by expertised BNs, respectively, as we stated in the 2nd paragraph of Page 6.
>
>
> > Summary of The Review:
> >
> > And the evaluation could include more details regarding training setups and efficiency.
>
>
> To show the training efficiency of our method, we add illustrations of the training process in the Appendix. B.1 and plot the convergence curves in Fig. 15.
> Thanks to the parallel training of multiple base models, our method (customizable for x0.125, x0.25, x0.5, and x1 nets) is much more efficient than SHeteroFL and only slightly slower than the FedAvg with x0.125 nets.
> In contrast, SHeteroFL suffers from the sequential training of models of different widths, resulting in large time complexity.

---

> ### Author Response · Authors · 2021-11-21
> **Response to Reviewer Yvwo on detailed comments**
>
> > Detail Comments:
> >
> > 1. The notations in the three algorithms lack necessary explanation.
>
> The notations are summarized in Alg. 3 due to the space limitation.
> To avoid changing the algorithm indexes, we will reorder the three algorithms after rebuttal.
>
> > 2. Regarding the second contribution stated in Sec. 1, it is better to present direct evidence of efficiency in storage, model loading and inference. For example, the time to load the model, the inference time and latency. The existing evaluations like #Params and FLOPs are indirect metrics
>
> We follow the common practice of previous customizable networks, e.g., (Yu et al., 2019; Wang et al., 2020; Huang et al., 2018; Wang et al., 2018) to evaluate the efficiency in terms of parameter numbers and FLOPs.
> The major reason is that parameter numbers and FLOPs are fair and stable measurement invariant to a dynamic testing environment.
> In contrast, even if testing hardware is specified, it could still be hard to reproduce the testing results of time measurement because of uncertain running conditions.
>
>
> > 3. The two considerations listed in Sec. 3 are somehow similar. It is better to clarify the difference between heterogeneous computational budgets and imbalance computation budgets.
>
> The imbalance computation budgets can be viewed as a special case of heterogeneous computation budgets.
> When more clients are incapable of training wider models, the customization will become harder.
> Therefore, with the task, we can better distinguish different methods and demonstrate the challenges of the proposed problem setting.
>
> > 4. The authors discuss their solutions on customising robustness as it is quite essential in federated learning. I would like to know how this idea can be adapted to other aspects like as mentioned in Sec. 3 neural -architecture search and fairness.
>
> If we train base models with different architectures or fairness levels (by regularizations), we can mix them adaptively for different in-situ customizations for inference efficiency-accuracy trade-off and fairness-accuracy trade-off.
>
> > 1. I am a little confused by the illustration of Fig. 5. So there are two BN in the training process, each of which handles noised/clean data. And in the inference process, the results are aggregated based on the coefficient. How are the noised and clean data differentiated during training? Are the label input together with the training data?
>
> During the training, given a batch of clean samples, we promptly synthesize noise samples from the clean samples by adversarially attacking (see the maximization in Eq. (1)).
> At inference time, the noise status of the input sample is agnostic to the model.
> Therefore, we input the noise-unaware sample into both batch-normalizations and then combine the outputs of BNs into the next layer.
>
> > 1. Regarding the evaluation setup, it is recommended to have more details on the clients' budget, e.g. the distribution of client budgets. Are there more clients with a large budget? I suppose the distribution will impact the ultimate accuracy performance.
>
> We explicitly define the budget distribution as $R_k = (1/2)^{\lceil 4 k/K \rceil}$ our problem setting (Sec. 3) and use it for the whole paper including the evaluation.
> In the distribution, there are 25% clients with large budgets (as defined by capability of training a x1 net), and 75% clients with smalles budgets (budget smaller than x1 net). We provide more experiments in Appendix B.3 where we increase the portion of budget-sufficient clients and visualize the distribution of budgets in groups (see Fig. 16).
>
>
> > 1. Some minor questions. In Fig. 9(a), why is there only one data point of FedAvg+OAT? Is it due to the divergence in training? In Fig. 7, why are there outliers? It is better to have some general description the baseline methods (i.e. OAT in-situ method) in the evaluation section. I am curious about the condition variable. So what is the ultimate setting in the evaluation section?
>
> We tuned the experiment but the OAT does not converge when combined with  FedAvg. As we have explained in Sec. 5.2 (the 2nd paragraph), the convergence issue of OAT may be a result of heterogeneous objectives and over-parameterization by conditional layers.

---

### Author Response · Authors · 2021-11-23
**Follow-up on rebuttal and a kind reminder**

We want to thank all the reviewers for the constructive suggestions and thoughtful reviews, which are valuable to improving our paper. As a follow-up on our rebuttal, we would like to kindly remind the reviewers that the close date of the discussion is approaching. We hope to use this open response window to discuss the paper, answer follow-up questions, and improve the quality of our paper. Have you gotten a chance to read our responses below, in which we tried our best to address your concerns? We want to make sure that the reviewers found our responses solid and convincing. And we would be more than happy to provide more information or clarification.

---

### Author Response · Authors · 2021-11-29
**A summarized response to all reviewers and the AC panel**

Dear reviewers and AC panel,

We appreciate all reviewers for reviewing our paper and leaving valuable comments! We are grateful that the merits of our work are acknowledged by Reviewer Yvwo, and 82wJ.

With all respect, we think some comments from Reviewer 7P9j and emNn are biased due to some misunderstandings of our paper.
We have tried our best to address his/her concerns but we still have not gotten further comments from Reviewer 7P9j until now (the last day of the discussion period).
Due to the situation, we would like to bring the major concerns and our summarized responses to the AC's attention.

1. **Contributions**. Reviewer 7P9j vaguely mentions that our contributions are not well explained. We tried to re-iterate our contributions and expect to get more specific comments but not until now. We want to refer to Reviewer 82wJ comments where our contributions are well understood and summarized: ‘Addressing data and resource heterogeneity is definitely one of the important topics of FL.’; ‘The main technical idea, … seems like a straightforward and reasonable solution to the problem.’ and ‘ jointly training both accurate and robust models while keeping all but BN layers shared sounds like an interesting idea’.
2. **Assumptions on the existence of budget-limited clients.** Though Reviewer 7P9j provide a negative example that Raspberry Pi 4 can afford training ResNet18, we argue that there are old generations of cell phones that cannot afford such training [1], for instance, LG Optimus L7 due to limited memory, or Asus Memo Pad 7 for large latency. Therefore, we don’t think the reviewer’s concern is well-supported.
3. **The advantages of Split-Mix for CIFAR10 and DomainNet are marginal**. The conclusion from Reviewer 7P9j is based on an unfair comparison between budget-constrained federated training and unconstrained ones.
4. **Other types of client heterogeneity** are included in the revision, including the non-i.i.d class distributions (in Appendix B.2) and a variety of budget distributions (Appendix B.3). The concern on the exponential budget distribution was raised by Reviewer 82wJ, as well. Though other reviewers did not further comment on this, we appreciate that Reviewer 82wJ is satisfied with our response to the problem.
5. **Empirical justifications for hypotheses.** Though we provide theoretic analysis and some empirical examples to motivate our methods, we follow the suggestions from Reviewer emNn to provide more references or experiments to demonstrate that: jointly training multiple base models (e.g., 4 x0.25 nets) cost fewer budgets than a single width-comparable network (e.g., 1 x1 net); minor adversarial noise can improve robustness in data-limited cases; experiments with larger datasets. Though these empirical results could solidify our work, our theoretical analysis is general regardless of hardware environments.


[1] Ignatov, A., et al. (2018). AI Benchmark: Running Deep Neural Networks on Android Smartphones. CVPR.

---

### Decision · Program_Chairs · 2022-01-20

**Decision:**

Accept (Poster)

**Comment:**

This paper proposes a federated learning (FL) scheme that is suitable for clients/devices with heterogeneous resources. The scheme Split-Mix trains multiple models of different sizes and adversarial-robustness levels, which are tailored to the budgets of the individual device. Empirical results show encouraging results.

It is clear that FL will have to work with clients with diverse resources, a point that is appreciated. Indeed, it is anticipated that widely-dispersed inference will have to deal with a highly-heterogeneous mix of clients. The study is quite thorough. One aspect that is not convincing in the experiments is the budgets being exponentially distributed: having a strong concentration around a mean (with something like a Gaussian tail), or a power-law distribution, would be more suitable.